# Cherries and Blueberries-Based Beverages: Functional Foods with Antidiabetic and Immune Booster Properties

**DOI:** 10.3390/molecules27103294

**Published:** 2022-05-20

**Authors:** Ana C. Gonçalves, Ana R. Nunes, José D. Flores-Félix, Gilberto Alves, Luís R. Silva

**Affiliations:** 1CICS-UBI—Health Sciences Research Centre, University of Beira Interior, Av. Infante D. Henrique, 6200-506 Covilhã, Portugal; anacarolinagoncalves@sapo.pt (A.C.G.); araqueln@gmail.com (A.R.N.); jdflores@usal.es (J.D.F.-F.); gilberto@fcsaude.ubi.pt (G.A.); 2CIBIT—Coimbra Institute for Biomedical Imaging and Translational Research, University of Coimbra, 3000-548 Coimbra, Portugal; 3CNC—Centre for Neuroscience and Cell Biology, Faculty of Medicine, University of Coimbra, 3000-548 Coimbra, Portugal; 4CPIRN-UDI-IPG—Center of Potential and Innovation of Natural Resources, Research Unit for Inland Development, Polytechnic Institute of Guarda, 6300-559 Guarda, Portugal

**Keywords:** functional foods, functional beverages, cherry, blueberry, health properties

## Abstract

Nowadays, it is largely accepted that the daily intake of fruits, vegetables, herbal products and derivatives is an added value in promoting human health, given their capacity to counteract oxidative stress markers and suppress uncontrolled pro-inflammatory responses. Given that, natural-based products seem to be a promising strategy to attenuate, or even mitigate, the development of chronic diseases, such as diabetes, and to boost the immune system. Among fruits, cherries and blueberries are nutrient-dense fruits that have been a target of many studies and interest given their richness in phenolic compounds and notable biological potential. In fact, research has already demonstrated that these fruits can be considered functional foods, and hence, their use in functional beverages, whose popularity is increasing worldwide, is not surprising and seem to be a promising and useful strategy. Therefore, the present review reinforces the idea that cherries and blueberries can be incorporated into new pharmaceutical products, smart foods, functional beverages, and nutraceuticals and be effective in preventing and/or treating diseases mediated by inflammatory mediators, reactive species, and free radicals.

## 1. Introduction

Food is any substance consumed capable of providing nutrients required for several functions, such as producing energy, supporting various metabolic activities, growing processes, and promoting wellness and a healthy status [1,2]. In recent decades, the demand for healthy foods and beverages has increased worldwide, mainly due to their nutritional values and health-promoting properties, demonstrating their ability to reduce the risk of oxidative stress-related disorders and others [3,4,5]. Given that, it is not surprising that the knowledge about the influence of nutrition on health and well-being has greatly increased, leading to the development of new and healthier foods, which are called functional foods [6].

The concept of functional foods was firstly described in ancient Vedic texts from India and was also an integral part of traditional Chinese medicine since early times [7]. In the 1980s, it was introduced in Japan, in the face of escalating processed foods that, in addition to their nutritional function, contained ingredients with specific bodily functions and beneficial physiological effects [7]. In 1984, after the increase in healthcare costs, an ad hoc group of the Ministry of Education, Science, and Culture in Japan launched a national project to explore the link between medical sciences and foods, and to legislate these products into Foods Of Specified Health Use (FOSHU) (Table 1) [8] given their biological potential [9]. To receive a FOSHU designation, manufacturers must complete an application that includes scientific evidence of the proposed medical or nutritional relationship, the proposed dose of the functional food, the safety of the food, and a description of the physical/chemical properties, experimental methods, and composition of the food [10].

The term “functional food” was first mentioned in 1993 in a scientific paper in Nature News Magazine entitled “*Japan explores the boundary between food and medicine*” [11]. There is no doubt about the interest of Japanese consumers in functional foods, and consequently, the growing awareness of functional products throughout the world. However, there is no consensus between Europe and the United States of America (USA) on a relatively concrete definition for functional foods, resulting in a variety of different terms: nutraceutical, designer food, pharmafood, and others, which contribute to increasing the confusion between professionals and consumers [12]. The USA prefers the term “nutraceutical”, while European experts decided to adopt the term “functional food” with a consensus definition within the FUFOSE (Functional Food Science in Europe) project (Table 2).

However, it is important to emphasize that functional foods must be foods and not medicines. Furthermore, the positive health effects should be achieved by consuming normal amounts of the functional food in question as part of a normal daily diet and should improve the quality of life. Interest in functional foods and beverages continues to grow, driven by ongoing research efforts to identify potential health-promoting properties and potential applications of nutraceuticals, resulting in increasing consumers’ interest in the role of foods and beverages in health and wellness. Within the emerging paradigm of functional foods, functional beverages can help to promote daily consumption of fruits and vegetables, as recommended by WHO/FAO [14], which recommends 400 g of edible fruits and vegetables per day. However, it is important to bear in mind that the consumption of juices intake does not equate to the ingestion of whole fruits and vegetables.

Given the mentioned, this review aims to summarize data on functional beverages made from cherries and blueberry and their various physiological functions, which allow them to be classified as functional foods for which health claims will be approved in the future.

## 2. Functional Foods Definition

Functional foods are used to increase certain physiological functions, and to prevent or even cure, diseases [15]. The concept of functional foods encompasses the idea that foods can play a role beyond gastronomic providing energy and nutrients [12]. Examples include conventional foods, fortified, enriched, or enhanced foods, and dietary supplements. For instance, the enrichment of traditional beverages, such as traditional beers, with phenolic compounds in order to increase, not only their flavor and attractivity for the consumer but also their nutritional values, have been increasing worldwide [16]. Therefore, functional foods provide essential nutrients in amounts beyond what is necessary for normal maintenance, growth, and development, and/or provide other biologically active components that have health benefits or desirable physiological effects. Functional foods are essentially a marketing term that is not recognized worldwide [7]. Several definitions of functional foods have been given (Table 3).

In the USA, there is no formal definition of functional foods, instead, the terms nutraceutical, dietary supplement, or medical foods, are used (Table 4), and therefore, functional foods cannot be regulated differently from other foods. Despite government authorities, national and international organizations have proposed their definitions of functional foods. The lack of a uniform definition in different countries has resulted in the unregulated publication of health claims for functional foods and a lack of clarity for scientists, governments, and consumers as to what “functional foods” actually are. There is an urgent need for researchers to rethink the meaning of functional food and agree on a new formal definition of functional foods that applies to all countries.

Although the terms “nutraceuticals” and “functional foods” are commonly used worldwide, there is no consensus on their meaning. Therefore, the Bureau of Nutritional Sciences, of the Food Directorate of Health Canada, has proposed the following definitions: a nutraceutical is a product isolated or purified from food, generally sold in medicinal forms (powders, pills, and other medicinal forms) that are not normally associated with food. A nutraceutical has been shown to have physiological benefits or provide protection against chronic diseases. Functional food has a possible appearance of a conventional food, and can be consumed as part of the usual diet. Beyond basic nutritional functions, it demonstrates several physiological benefits and/or the ability to reduce the risk of chronic disease [17]. According to regulation (functional food [17]), a functional food is a food with specific beneficial effects on one or more target functions in the body that go beyond basic nutritional functions and result in improved health status and well-being or a reduction in the risk of disease. It is consumed as part of a normal diet and is not used in the form of pills or capsules or any other form of dietary supplement [18]. 

Functional foods are clearly in a different category than nutraceuticals, pharmafood, or dietary supplements. They are considered food, not a pharmaceutical drug because they have health-promoting properties, which are usually disease-preventive, rather than therapeutic properties (Table 3 and Table 4).

**Table 3 molecules-27-03294-t003:** Several definitions of functional foods.

Reference	Definition
FOSHAN [19]	Foods for specified health use. The FOSHU can be foods that exhibit health effect, used as foods in a diet, and are in the form of foods, not as supplements
Health Canada, Ontario, Canada [20]	A functional food to be similar in appearance to conventional food, to be consumed as part of the usual diet, to demonstrate physiologic benefits, and/or to reduce the risk of chronic disease beyond basic nutritional functions.
International Food Information Council, Washington, USA [21]	Foods or dietary components may provide a health benefit beyond basic nutrition.
International Life Sciences Institute of North America (ILSI North America) [22]	Foods that by physiologically active food components provide health benefits beyond basic nutrition.
Regulation (EC) No 1924/2006 [18]	Functional food is a food with certain beneficial effects on one or more target functions in the body beyond the basic nutritional effects with a result of the improved health state and well-being or reduction of risk of diseases. It is consumed as a part of a normal diet and is not used in the form of a pill or capsule or any other form of dietary supplement.
[7]	A food product can be made functional by using any of the five approaches listed below: (1) Eliminating a component known to cause or identified as causing a deleterious effect when consumed (for example, an allergenic protein). (2) Increasing the concentration of a component naturally present in food to a point at which it will induce predicted effects (for example, fortification with a micronutrient to reach a daily intake higher than the recommended daily intake). (3) Adding a component that is not normally present in most foods and is not necessarily a macronutrient or a micronutrient, but for which beneficial effects have been shown (for example, non-vitamin antioxidant or prebiotic fructans). (4) Replacing a component, usually a macronutrient (for example, fats), intake of which is usually excessive and replacing it with a component for which beneficial effects have been shown (for example, modified starch). (5) Increasing bioavailability or stability of a component known to produce a functional effect or to reduce the disease-risk potential of the food.
Functional Food Center (FFC) [10]	Natural or processed foods that contain known or unknown biologically-active compounds; which, in defined, effective non-toxic amounts, provide a clinically proven and documented health benefit for the prevention, management, or treatment of chronic disease. In this definition, first functional foods can be natural or processed. Second, bioactive compounds, which are considered to be the source of the functionality of the foods, are secondary metabolites that occur in food usually in small amounts that act synergistically to benefit health. Specifically, bioactive compounds may exert antioxidant, cardio-protective and chemo-preventive effects.
Food and Nutrition Board (FNB) of the National Academy of Sciences, Washington, USA) [23]	Functional food is one that encompasses potentially healthful products, including any modified food or food ingredient that may provide a health benefit beyond that of the traditional nutrient it contains.

**Table 4 molecules-27-03294-t004:** Terms linked with functional foods.

**Bioactive Compounds:** The Office of Dietary Supplements at the NIH has defined bioactive compounds as constituents in foods or dietary supplements, other than those needed to meet basic human nutritional needs, which are responsible for changes in health status [24].
**Dietary Supplements:** Dietary supplements mean foodstuffs, the purpose of which is to supplement the normal diet, and which are concentrated sources of nutrients or other substances with a nutritional or physiological effect, alone or in combination, marketed in dose form, namely forms such as capsules, pastilles, tablets, pills and other similar forms, and sachets of powder, ampoules of liquids, drop dispensing bottles, and other similar forms of liquids and powders designed to be taken in measured small unit quantities [25].
**Functional Ingredients:** Functional ingredients are a diverse group of compounds; health benefits have been attributed, for example, to allyl compounds found in garlic, carotenoids, and flavonoids, found in fruits and vegetables, glucosinolates, found in cruciferous vegetables, hypericin and pseudohypericin found in St. John’s wort, peptides such as epidermal growth factor, opioid peptides, and lactoferrin, found in milk, and arachidonic and docosahexaenoic acids, found in human milk and derived for use in infant formulas from various algal, bacteria, and fish sources. Functional ingredients can be marketed as part of dietary supplements, food additives, or generally recognized as safe (GRAS) ingredients included in functional foods [23].
**Medical Foods:** A Medical Food is a food that is “formulated to be consumed or administered under the supervision of a physician and which is intended for the specific dietary management of a disease or condition for which distinctive nutritional requirements are established by medical evaluation [26].
**Natural health products:** Natural health products (NHPs) include homoeopathic preparations, substances used in traditional medicine, a mineral or trace element, a vitamin, an amino acid, an essential fatty acid, or other botanical-, animal-, or microorganism-derived substance [27].
**Nutraceutical:** The term nutraceutical is a substance that may be considered a food or part of a food that provides medical or health benefits, encompassing prevention and treatment of disease. Products as diverse as isolated nutrients, dietary supplements, and diets, to genetically engineered “designer” foods, herbal products, and processed foods (cereals, soups, beverages) may be included under the umbrella of nutraceuticals [28].

Together with fruits, vegetables, nuts, seeds, and grains, some beverages can also be considered functional foods (Table 5) [12,29,30]. For example, oats contain beta-glucan, a dietary fiber associated with reducing inflammation, boosting immune function, and improving heart health [31]. In addition, fruits and vegetables are packed with antioxidants, which are beneficial compounds that protect against disease [7,29]. This category also includes foods enriched with vitamins, minerals, fatty acids, probiotics, and prebiotics (fiber, fructooligosaccharides, inulin, lactulose, and sugar alcohols) [12].

Several dairy products have also been explored, such as yogurts with live cultures and lactose-free cheeses. The addition of margarine is another commonly available functional food product. Examples of functional beverages are energy drinks and those enriched with vitamins and minerals or lactose-free milk (Table 5) [7].

In conclusion, there is a consensus about the term functional which is used to enhance certain physiological functions in order or even cure medical conditions; however, some control exists about the fact that capsules, pills, and powders might be included [15]. A functional food can be (i) a natural food, (ii) a food to which a component has been added, (iii) a food from which a component has been removed, (iv) a food where one or more components has been modified, (v) a food in which the bioavailability has been modified, or (vi) any combination of these [7]. There is no EU legislation on functional foods, so this definition has legal force. It is only a current working definition: functional foods are not pills, capsules, or any form of food supplement, but in any case, they must retain the character of a food, and their consumption must be part of a normal diet.

## 3. Fruits and Beverages as Functional Foods

The global consumption of fruit beverages reached 95.69 billion liters in 2018, representing 0.78% less compared to 2017, accounting for 37.23 billion juice drinks, 10.92 billion liters for nectar, and 30.55 billion for powdered and concentrated juice [32], revealing its great importance in the world economy. Consumers demonstrated an increased interest in juices with innovative and functional juice ingredients that help improve health. With the pandemic coronavirus in 2020, demand for juice in Europe rapidly increased. It is common to see orange juice consumption increase during the flu season in Europe. However, COVID-19 completely changed consumption patterns. Until 2019, European juice consumption decreased at an annual rate of 1% for almost a decade. But in 2020/2021, consumption increased again in several European markets. After the COVID-19 pandemic began, consumers regained some interest in citrus juices and red fruits. They also became more interested in functional ingredients in drinks [33].

Functional beverages occupied over half (USD 99 billion) [13] of the total market value (USD 168 billion) of functional foods in 2019 [14], with approximately 1/3 of the market value (USD 36 billion) being contributed by the Asia Pacific region [13].

The growth of functional beverages and novel ingredients has led to improved legislation (Regulation No. 258/97) [34] from the European Commission (EC), which now requires the submission of a comprehensive safety document before novel ingredients can be used in foods. The EC is willing to grant approval and support the industry provided their development is based on scientific evidence and validated in human populations [33]. Within the emerging paradigm of functional foods, functional beverages may help increase the consumption of fruits and vegetables to restore the balance between recommendations and actual consumption, although consumption of juices is not equivalent to the consumption of whole fruits or vegetables.

In 2007, the regulation regarding nutrition and health claims made on foods was introduced in the European Union. This regulation provides opportunities for the use of health claims on foods in Europe, including claims to reduce the risk of disease. Nutrition and health claims must be based on and substantiated by generally accepted scientific evidence; EFSA requires RCT (Randomized Control to demonstrate the beneficial physiological effects on a healthy population [35,36].

Consumers around the world are looking for healthier, more natural, and functional products, looking for relaxation, energy, performance, and memory, as well as the addition of exotic ingredients and vegetables [29]. The consumption of low-sugar options has already been consolidated in the market by natural substitutes [32]. Beverages are commonly used to deliver high concentrations of functional ingredients (e.g., sports and performance drinks, ready-to-drink teas, vitamin-enriched water, soy, and energy beverages) [29].

Functional beverages have been reported as the most active and popular among consumers, taking into account meeting consumer demands for desirable nutrients and bioactive compounds, easiness in distribution and storage, size, shape, and appearance [37]. Beverages have been used habitually to deliver high concentrations of functional ingredients, associated with their easy delivery and human body need. Beverages represent an appropriate solution for the dissolution of functional ingredients, but also a convenient and widely accepted method of consumption. The processing of beverages can contribute to some sensory barriers (e.g., bitter taste, grainy texture, etc.), and they provide a proper method of ingestion [29]. The different types of commercially available beverages could be grouped into: (i) dairy-based beverages including probiotics and minerals/ω-3 enriched drinks, (ii) vegetable and fruit beverages, and (iii) sports and energy drinks [37].

As far as the study of fruit juice is concerned, two major groups have been focused on: (i) juices high in antioxidants, or (ii) juices relatively low in antioxidants, but widely consumed by the general public. However, there are also mixed juices with high antioxidant activity that are consumed in relatively large quantities [29]. The first group of juices includes pomegranate, cranberry, and blueberry, as well as other dark fruits, such as cherry and blackcurrant, which have higher levels of phenolic compounds (e.g., phenolic acids, flavonoids, anthocyanins, and tannins) [38,39,40] In the other category, research focused on orange, grape, and apple juices, which contain mainly hydroxycinnamic acids and vitamins [29,41,42].

Berries, such as blueberries and sweet cherries, are usually consumed as fresh fruits, however, but various technological products are also widely available. They are usually processed into juices, concentrate, and jams/purees; additionally, their oils can be extracted from seeds [30]. Interest in beverages has increased significantly in recent years. Studies have demonstrated more beneficial effects of berries phytochemicals, which has led to an increase in consumption associated with increased health awareness among consumers [30,39,40,43,44].

Red fruits such as berries are an important component of a healthy diet due to their high content of phenolic acids and flavonoids, especially anthocyanins [30,40,43,44]. Blueberries (*Vaccinium* spp.) and cherries (*Prunus* spp.) are considered one of the five healthy foods certified by the International Food and Agriculture Organization (FAO) because they are rich in phenolic compounds, anthocyanins, and other nutrients [45]. Their consumption has increased in recent years, partially due to the health benefits attributed to their phenolic content. Blueberries are composed of high levels of anthocyanins, flavonols, and flavan-3-ols, as well as benzoic and cinnamic acids [43,46,47]. Several health benefits were reported for blueberries and cherries given their capacity to offer protection against metabolic disorders thanks to their remarkable antioxidant, anti-inflammatory anti-diabetic properties [48,49].

## 4. An Overview Regarding Cherry and Blueberry Phenolic Compounds

In current days, a crescent interest in natural antioxidant molecules has been increasing worldwide, once, contrary to synthetic antioxidants, it is believed that their continuous intake does not present, or presents fewer, undesirable side effects [50]. In line with the mentioned, it is not surprising that cherries and blueberries, particularly *Prunus cerasus* (tart cherries) and *P. avium* (sweet cherries), and *Vaccinium corymbosum* (highbush blueberries) and *Vaccinium ashei* (rabbiteye blueberries), are a target of much research [40,51,52,53,54]. Indeed, they are considered super red fruits, given their fullness in several phenolic compounds (total phenolic compounds (TPC) of around 275.3–484.1 for tart cherries [55,56], 28.3–493.6 for sweet cherries [53,57,58] and 2.7–585.3 and 390–2625 mg gallic acid equivalents (GAE) per 100 g fresh weight (fw) for highbush and rabbiteye blueberries, respectively [53,55,56,57,58,59,60,61,62,63,64,65,66]) (Table 6). Of course, these amounts depend on many factors, including genotype, local origin, methods of cultivation, and maturation stage, processing, and storage conditions, among others [40,52].

Phenolic compounds are secondary plant metabolites produced via shikimic and acetic acids, and phenylpropanoid and flavonoid pathways, to offer protection to plants against biotic and abiotic factors [67]. Given that, it is not surprising that their intake is also an added value [65]. Among phenolic subclasses, it is important to underline the presence of anthocyanins, i.e., the glycosides of anthocyanidins (total anthocyanin concentrations (TAC) of around 21.0–295 for tart cherries [54,68], 3.7–98.4 [53,54] for sweet cherries and 34.5–552.2 and 69.97–378.31 mg cyanidin 3-*O*-glucoside equivalents per 100 g fw for highbush blueberries and rabbiteye blueberries, respectively [53,54,64,68,69,70,71]) (Table 6). These molecules are considered the key responsible for the organoleptic properties and biological potential demonstrated by these fruits given their multiple hydroxyl groups, and also owing to their electron deficiency, which confers to them the easy capacity to neutralize and/or reduce free radicals and reactive species. In most fruits, including red fruits, anthocyanins are commonly found conjugated with arabinose, galactose, glucose, or rutinoside sugar in order to be more stable (Figure 1). Among them, cyanidin 3-*O*-glucosyl-rutinoside (89.0–227.66 mg per 100 g fw), cyanidin 3-*O*-rutinoside (1.76–74.7 per 100 g fw), cyanidin 3-*O*-sophoroside (0.13–10.0 mg per 100 g fw), cyanidin 3-*O*-glucoside (0.01–1.03 mg per 100 g fw), and cyanidin aglycone (31–6.64 mg per 100 g fw) are the main anthocyanins reported in tart cherries [54,70,72,73]. Additionally, trace amounts of peonidin 3-*O*-glucoside, cyanidin 3-*O*-xylosylrutinoside, cyanidin 3-*O*-galactoside, delphinidin 3-*O*-rutinoside, and delphinidin, malvidin, peonidin, and pelargonidin aglycones are also detected [70,72,74,75]. On the other hand, sweet cherry fruits present higher amounts of cyanidin 3-*O*-rutinoside, which represents around 90.0% of total anthocyanins found, and from 42.5 to 68.6% of total phenolic compounds detected, followed by cyanidin 3-*O*-glucoside, at levels varying between 0.20 and 389.9, and from 0.0 to 142.03 mg per 100 g fw, respectively [40,43,76,77], and by vestigial amounts of pelargonidin and delphinidin 3-*O*-rutinoside, peonidin, and malvidin derivatives, and cyanidin and malvidin aglycones [40,43,53,54,78,79]. Regarding highbush blueberries, they are richer in malvidin 3-*O*-galactoside (12.11–67.45 mg per 100 g fw), peonidin 3-*O*-glucoside (12–54.37 mg/100 g fw), delphinidin 3-*O*-glucoside (1.21–53.62 mg per 100 g fw), delphinidin 3-*O*-galactoside (2.29–53.29 mg per 100 g fw), delphinidin 3-*O*-arabinoside (1.66–41.07 mg per 100 g fw), malvidin 3-*O*-glucoside (0.68–34.75 mg per 100 g fw), petunidin 3-*O*-galactoside (2.57–28.54 mg per 100 g fw), and petunidin 3-*O*-glucoside (0.67–25.14 mg per 100 g fw) [59,80,81,82]. Additionally, they also contain trace amounts of cyanidin 3-*O*-glucoside, cyanidin 3-*O*-arabinoside, cyanidin 3-*O*-galactoside, cyanidin 3-*O*-hexoside, petunidin 3-*O*-arabinoside, malvidin 3-*O*-arabinoside, malvidin-3-(6″-acetyl-galactoside), malvidin-3-(6″-acetyl) glucoside, malvidin 3-*O*-xyloside, peonidin-3-*O*-pentose, peonidin 3-*O*-galactoside, and cyanidin, delphinidin, cyanidin, peonidin, malvidin, and petunidin aglycones [59,81,82,83]. On the other hand, peonidin 3-*O*-glucoside is the most abundant in rabbiteye blueberries, followed by malvidin 3-*O*-glucoside, malvidin 3-*O*-arabinoside, and delphinidin 3-*O*-galactoside [84,85].

As well as other food matrices, these super red fruits also contain other phenolics in their composition capable of reinforcing their putative biological properties, namely phenolic acids, flavonols, flavan-3-ols, and flavones [40,67,81,101,105]. Regarding hydroxybenzoic acids (Figure 2A), none are reported yet in tart cherries, however, *ρ*-hydroxybenzoic, gallic, and protocatechuic acids are found in the sweet ones, at levels of 10.3–19.1, 0.0018–1064 and 0.054–3.28 mg per 100 g fw, respectively [40,76,78,96,106]. In addition, residual amounts of protocatechuic acid glycoside and protocatechuoyl hexose are also reported in sweet cherries [76]. Comparatively to cherries, ellagic acid was, until now, only detected in both blueberries, at amounts varying from 0.75–6.65 to 0.0–19.25 mg per 100 g fw for highbush and rabbiteye, respectively [60,64]. Vanillic acid was only found in the highbush fruits (0.011–0.027 mg per 100 g fw) [94], which in turn, also present higher levels of *ρ*-hydroxybenzoic, protocatechuic, and syringic acids than cherries (0.054–59.78, 5.22–41.45, and 0.034–9.95 mg per 100 g fw, respectively) [63,64,94].

Concerning the presence of hydroxycinnamic acids (Figure 2B), some differences are also noticed. For instance, *ρ*-coumaric acid is the most abundant in tart cherries (0.89–5.69 mg per 100g fw), followed by 3-caffeoylquinic and 5-caffeoylquinic acids (values ranging from 0.58 and 60.33, and 5.24.27.79 mg per 100 g fw, respectively) [54,56,74]. These phenolic acids, together with 3-coumaroylquinic acid, are also the predominant ones in sweet cherries [40,43,54,57,76,96]. As well as cherries, both highbush and rabbiteye blueberries present considerable amounts of 5-caffeoylquinic and *ρ*-coumaric acids [60,64,94]. Nevertheless, sinapic and caftaric acids were only detected in highbush blueberries, at amounts fluctuating between 0.005 and 0.11, and around 4.71 mg per 100 g fw, respectively [63,77,94].

Regarding flavonols (Figure 3), quercetin 3-*O*-rutinoside and kaempferol 3-*O*-rutinoside are the most prevalent in tart cherries, at levels ranging from 0.84 to 7.63, and between 0.30 and 1.29 mg per 100 g fw, respectively [54,65,74,99]. Additionally, Sokół-Łȩtowska and coworkers [74] revealed the presence of isorhamnetin 3-*O*-rutinoside in Polish sour cherries (0.0 to 5.37 mg per 100 g fw). As well as tart cherries, the sweet ones also present considerable levels of quercetin 3-*O*-rutinoside and kaempferol 3-*O*-rutinoside (0.78–51.97 and 0.90–8.13 mg per 100 g fw, respectively), and also higher amounts of quercetin *O*-glucoside-*O*-rutinoside II (3.67–132.7 mg per 100 g fw) [43,65,76,98]. On the other hand, quercetin (0.29–21.48 mg per 100 g fw), kaempferol (0.061–19.65 mg per 100 g fw) and myricetin (6.72–6.98 mg per 100 g fw) aglycones, and quercetin 3-*O*-glucoside (0.90–34.64 mg per 100 g fw) are commonly the most reported in highbush blueberry cultivars [51,59,64,80]. Relatively to rabbiteye blueberries, they are also rich in myricetin and quercetin aglycones (6.72–6.98 and 0.046–9.97 mg per 100 g fw, respectively) [51,60,64]. Additionally, they also present various derivatives of myricetin, quercetin, laricitrin and syringetin, demonstrating the presence of myricetin 3-*O*-rhamnoside, myricetin 3-*O*-glucuronide and myricetin 3-*O*-glucoside, quercetin 3-*O*-glucuronide, quercetin 3-*O*-hexoside, quercetin 3-*O*-rutinoside, quercetin 3-*O*-rhamnoside, laricitrin 3-*O*-glucuronide, and syringetin 3-*O*-glucoside [51,63,100].

Focusing on flavan-3-ols (Figure 4A), tart cherry contains (−)-epicatechin (0.0–28.22 mg per 100 g fw), procyanidin B1 (0.0–27.69 mg per 100 g fw), and procyanidin C1 (0.0–8.60 mg per 100 g fw) in their constitution [74]. (−)-Epicatechin is also one of the most predominant flavan-3-ols reported in sweet cherries (0.23–397.19 mg per 100 g fw), followed by (+)-catechin (0.13–84.34 mg per 100 g fw) and procyanidin dimer type 2 (1.59–26.47 mg per 100 g fw) [40,57,76,77,96]. Relative to blueberries, considerable amounts of (+)-catechin and (−)-epicatechin were detected in both highbush and rabbiteye blueberries [60,64,94].

Relatively to other non-colored phenolics, vestigial amounts of flavone luteolin, flavanone naringenin hexoside, and flavanonols taxifolin 3-*O*-rutinoside and taxifolin *O*-deoxyhexosylhexoside were already reported in sweet cherries [76,90,96]. On the other hand, naringenin aglycone, flavone apigenin 8-*C*-glucoside, coumarin aesculin, and chalcone phlorizin were found in highbush blueberries [94,96] (Figure 4B–E).

Without surprises, and although there is still a lack regarding the red-fruit juices analysis, cherry and blueberry juices also contain substantial amounts of phenolics, anthocyanins being the most found (Table 6). Therefore, and in line with the fruit, tart cherry juice is also rich in cyanidin 3-*O*-glucosyl-rutinoside (92.86–441.11 mg/L), cyanidin 3-*O*-sophoroside (1.62–292.21 mg/L), and cyanidin 3-*O*-rutinoside (0.38–85.5 mg/L) [87,88]. On the other hand, cyanidin 3-*O*-rutinoside and cyanidin 3-*O*-glucoside are the main ones in sweet cherry juice, with values ranging from 104.0 to 210.0 mg/L, and between 22.0 and 37.0 mg/L, respectively [89,97]. Regarding blueberry juices, they contain higher levels of delphinidin 3-*O*-galactoside (0.14–223.0 mg/L), petunidin 3-*O*-glucoside (7.70–365.0 mg/L), delphinidin 3-*O*-arabinoside (0.67–134.0 mg/L), malvidin 3-*O*-arabinoside (4.60–73.0 mg/L), and malvidin 3-*O*-glucoside (6.25–271.0 mg/L) [66,92,93,107,108]. Concerning non-colored phenolic compounds, the most found in tart cherry juices include 5-caffeoylquinic acid (28.30–995.0 mg/L), 3-coumarolyquinic acid (91.0–555.0 mg/L), 3-caffeoylquinic acid (82.0–183.0 mg/L), ferulic acid (1.14–1.27 mg/L), quercetin aglycone (184.0–739.0 mg/L), quercetin 3-*O*-rutinoside (4.0–53.80 mg/L), and (−)-epicatechin (13.60–369.0 mg/L) [56,87,88,100,109]. On the other hand, sweet cherry juices are mainly composed of 3-caffeoylquinic acid (24.77–37.78 mg/L), syringic acid (6.64–14.46 mg/L), gallic acid (0.0–6.55 mg/L), and quercetin 3-*O*-rutinoside (0.0–4.74 mg/L) [97,103]. Finally, blueberry juices are richer in luteolin 7-*O*-glucoside (ca., 102 mg/L) and quercetin 3-*O*-rutinoside (ca., 65 mg/L) [45,103,107]. Additionally, they present residual amounts of *ρ*-hydroxybenzoic acid, vanillic acid, 5-caffeoylquinic acid, caffeic acid, syringic acid, ferulic acid, and quercetin 3-*O*-glucoside [95].

To increase the nutritional values of berry fruits and derivatives, including juices, several efforts have been conducted. For instance, it was already reported that advanced technologies, including ultrasonic-assisted extraction, pulsed electric fields, microwave, and high hydrostatic pressure are more effective in extracting phenolic amounts from cherries than conventional solvent extraction (only involving homogenization) [56,89,96]. The coating of cherries with chitosan also seems to be a promising strategy, being able to preserve their firmness and enhance phenolic levels [110]. Additionally, Özen and colleagues reported that sour cherry produced using the concentrate juice present higher amounts of phenolic and volatile compounds [109], whereas Filannino et al. [111] verified that the fermentation of cherry juice with *Lactobacillus* spp. enriches its juice with phenolic derivatives with high bioavailability and biological activity. Similar data were obtained by Ricci and coworkers [112] who proved that the fermentation of cherry juice with lactic acid bacteria can enhance aromatic compounds and phenolic acids levels. Focusing on bilberries, it was reported that the exposure of blueberry plants to 100 μM of cadmium and aluminium for 14 days is effective in increasing phenolic compounds, duplicating their amounts comparatively with samples without any treatment [113]. Furthermore, a recent study also reported that dried blueberries at 50 °C can increase TPC and TAC values of their juice by 199.11 and 166.79%, respectively [66]. Furthermore, Zhu and colleagues [107] a reported that microchip-pulsed electric field (350 V, 0.15 ms, 7 mL/min) seems to be a promising innovative way of processing fresh blueberry juice, being capable of extending its shelf life, protecting its organoleptic properties (color and flavor) and enhancing its TPC values after 30 days of storage (+126.72%). It had been also reported that the acidification of blueberry juice (pH 2.1), combined with refrigerated storage (4 °C) and followed by pasteurization, is capable of preserving more anthocyanins (+56% anthocyanins than juices storage at 25 °C, and +12% more anthocyanins than juices at pH 2.5) [114]. More recently, Martín-Gómez and coworkers [115] mixed blueberry and grape musts with four yeast strains of *Saccharomyces cerevisiae* (X5 (CECT131015), M05 Mead, QA23 commercial yeast and MP061 Viniferm EP 837) in a 1:1 ratio, and verified that the fermentation with M05 Mead yeast strain was the most appropriate, being effective in preventing color losses and increasing anthocyanins amounts (+2.10% more than the initial sample without fermentation).

In addition, it is always important to take into account that cherries and blueberries are perishable fruits and hence, it is very important to control temperature and reduce vibration during transportation, and use adequate storage conditions and processing methods, to preserve their nutritional value and life [116].

## 5. Absorption, Digestion, Metabolism, and Bioavailability of Phenolic Compounds

Although understanding the absorption, digestion, metabolism, and bioavailability of phenolics is also not so easy, this is an essential tool to help in explaining the biological potential of these metabolites [117,118]. Until a very short time ago, it is believed that phenolics had lower absorption rates and were rapidly excreted; however, recent studies already reported that phenolics suffer an extensive metabolization along the gastrointestinal tract, and therefore, the absorption rates are higher than expectable, and some of them can be almost totally assimilated [119]. In a general way, most phenolics reach the maximum plasma concentration 1.5–2 h after intake [120]. Among phenolic subclasses, isoflavones seems to possess the highest absorption rates (33.0–100.0%), followed by hydroxycinnamic acids (8.0–72.0%), anthocyanins (2.40–55.0%), flavonols and flavones (12.0–41.0%), flavanones (11.0–16.0%), lignans (2.70–12.20%), and flavan-3-ols (2.0–8.0%) [121,122]. On the other hand, their excretion content is estimated to be around 30%, varying between 12 h (flavanols) and 24 h (anthocyanins and hydroxycinnamic acids derivatives) [123,124]. Of course, these percentages are relative and highly variable, being largely influenced by several features that can affect the transport, solubility, and permeability of each phenolic, such as degree of polymerization, molecular size, chemical structure, glycosylation pattern, pre-systemic metabolism, resistance to pH variations along the gastrointestinal tract and facility to release from food matrix [125,126]. Additionally, they also depend on factors responsible to influence phenolic content, such as agronomic practices, maturity stage, food matrix maturity and processing, and whether phenolics are ingested after fasting periods or not, and/or are alone or accompanied by other compounds, and also on gender, age, dietary habits, lifestyle, and genetic, and physiological and pathological states, as the possible existence of food intolerances and intestinal flora of each individual [127,128].

To circumvent these vicissitudes and increase the absorption and consequent bioavailability of phenolics, several studies have been conducted, and their encapsulation with proteins (e.g., gelatine, soy proteins, dairy derivatives), lipids (e.g., waxes and emulsifiers), natural gums (gum Arabic and alginates), and/or carbohydrates (maltodextrins and cellulose derivatives) seems to be a useful and promising tool [117].

Everything starts in the mouth with mastication, where, after phenolics’ intake, saliva enzymes initiate the degradation of glycosides [129] (Figure 5). Subsequently, they are conducted to the stomach [125]. Here, phenolics, namely simple aglycones (e.g., isoflavones) and phenolic acids (e.g., gallic acid) can be directly absorbed by bilitranslocase or reach intestinal epithelial cells by passive diffusion given their high lipophilicity, or through the active sodium-dependent glucose cotransporter (polymers, glycosides, and esters), becoming available for absorption [130,131]. However, most of the compounds remain intact and are conducted to the small intestine to be hydrolyzed by lactase-phlorizin hydrolase (LPH) and cytosolic *β*-glucosidase (CBG) enzymes and/or undergo reactions of methylation, glucuronidation, and sulfation [101,126]. After that, aglycones are conducted to the liver via the portal vein and the non-absorbed ones go to the colon to be once more degraded and modified into simpler monomeric units, thanks to reactions of decarboxylation, demethylation, dihydroxylation, and hydrolysis carried out by bacterial esterases (e.g., α-rhamnosidases) to suffer a new attempt of absorption by the liver [132,133]. In fact, recent data reveal that most phenolics are only absorbed after undergoing colon metabolization (90.0–95.0%) [134]. Usually, anthocyanins and proanthocyanidins are metabolized from low molecular weight phenolic acids, flavones, and flavanones into hydroxyphenylpropionic acids, and polymeric phenolic compounds, flavonols to hydroxyphenylacetic acids, and flavanols into phenylvalerolactones and hydroxyphenylpropionic acids, and after, degraded into hydroxybenzoic acid derivatives [131,135,136]. Additionally, it had been recently reported that the cleavage of phenolic glycosidic bonds on the colon leads to the formation of short-chain fatty acids, and hence, promotes the diminution of pH values and creates favorable conditions for the proliferation of probiotic bacteria, such as *Actinobacteria, Bifidobacteria,* and *Lactobacilli*, which in turn, exert positive effects in the control of gastrointestinal and digestive disorders, allergies, eczema, and improvements in delicate cases of cardiovascular and neurological ailments [117,135].

In the liver, most phenolics are converted into the simpler and lower molecular weight that becomes accessible to reach the systemic circulation and be uptook by organs and tissues after being subjected to phase I and phase II biotransformation reactions, which include specific reactions of oxidation, hydrolysis, methylation, glucuronidation, and sulfation (phase I) and the action of sulfotransferases (SULTs), catechol-*O*-methyltransferases (COMTs), and UDP-glucuronosyltransferases (UGTs) (phase II), or to enhance phenolics’ bioavailability; in the small intestine and kidneys the phase II of metabolism also occurs [137,138]. Furthermore, the non-absorbed phenolics in the liver can return to the small intestine via bile, to be again metabolized [126]. The non-absorbed phenolics are eliminated feces and urine [118].

Nowadays, and considering the large health-promoting properties of these berries, some research focused on studying their bioavailability-related processes has been conducted. Regarding cherries, Mihaylova and coworkers [139] investigated the effect of in vitro digestion on the phytochemical content of sweet cherry juice and only detected chlorogenic acid after the digestion procedure, probably resulting from the anthocyanins’ metabolization. Additionally, these authors reported that cherry flavonoids were more resistant to the digestion process, revealing bioaccessibility percentages of around 67%. Furthermore, it was also documented that sour cherry phenolics were stable during gastric conditions, and around 59% of their content can be released into the bloodstream and reach tissues and organs [127]. Additionally, Kirakosyan et al. [134] detected higher values of cyanidin 3-glucosyl-rutinoside (2339 picograms/gram of tissue) and cyanidin 3-rutinoside 5-*β*-D-glucoside (916 picograms/gram) in the bladder and liver of rats, respectively. Focusing on blueberries, it was reported that their phenolics are absorbed and extensively metabolized by phase II enzymes and gut microbiota of humans, originating various metabolites that may be responsible for the beneficial effects observed after blueberry consumption. In total, 61 phenolic metabolites were quantified in the plasma at baseline, of which 43 increased after consumption of blueberries over 30 days. Among metabolites, benzoic and catechol derivatives represented more than 80% of the changes in the phenolic profile after 2 h of consumption, while benzoic and hippuric derivatives were the major compounds after 30 days of daily intake. The phenolic urinary excretion remained unchanged and no systemic toxicity was found [139]. Particularly, Zhong et al. [120] reported that anthocyanins and 3-chlorogenic acid peaked at their maximum 2 h after blueberry ingestion, whereas phase II metabolites, including glucuronide conjugates of peonidin, delphinidin, cyanidin, and petunidin, peaked at 2.6, 6.3, 7, and 8.8 h, respectively. Phenolic acid metabolites’ peak occurs between 0.5 and 24 h. More recently, Serra et al. [119] subjected their anthocyanin-rich extracts to an in vitro experiment capable of mimicking gastrointestinal digestion, and in the final experiment, they verified the maintenance of higher monoglycosides of cyanidin, malvidin, delphinidin, peonidin, and petunidin levels. Their TPC levels were 9000 mg/L and 850 mg/L GAE, before and after digestion, respectively. Within them, derivatives of malvidin were the most predominant, which is expected since they present fewer hydroxyl groups than other anthocyanins, and, hence, are more resistant to digestion degradation. These data are in accordance with a previous study, which revealed the presence of malvidin 3-*O*-glucoside in the plasma and urine of human volunteers after the intake of red grape juice, indicating that it is not significantly degraded, being absorbed in its glycosylated form [140]. Additionally, it has also been detected that protocatechuic and vanillic acids resulted mainly from the metabolization of cyanidin 3-*O*-glucoside [119]. Moreover, and as already mentioned, phenolics’ encapsulation seems to be an effective technique, given its capacity to enhance their stability. Given that, Wu and collaborators [117] decided to encapsulate blueberry phenolics with Arabic gum, maltodextrin, gelatin, and soy protein isolate and verified that the use of the last two wall materials can increase bioaccessibility anthocyanin levels and promote intestinal health, by increasing *Bacteroidaceae* levels, which in turn, leads to the production of acetic, propionic, and butyric acids. Syringic acid was detected and its presence is from the degradation of anthocyanins during colonic fermentation. Similar results were obtained by conjugating blueberry anthocyanins with bovine serum albumin [141].

## 6. Functional Properties of Cherries and Blueberries—Focus on Antidiabetic and Anti-Inflammatory Potential of Phenolic Compounds

Diabetes is a chronic metabolic disease affecting about 463 million people worldwide, with major economic and well-being impacts [142]. Hyperglycemia is responsible for structural and functional alterations in the body resulting from the increase in oxidative stress, the formation of advanced glycation end products (AGEs), inflammation, and protein glycation [143]. In the long term, diabetes can lead to severe complications such as myocardial infarction and stroke [144].

New therapeutic approaches, in addition to antidiabetic drugs and insulin therapy, are required. Beyond the side effects, conventional therapy is expensive and sometimes not accessible to all [143,145]. Thus, the research and development of novel formulations using natural products, rich in phenolic compounds and with antidiabetic properties, is a promising strategy for diabetes management [43,146,147]. In this context, functional foods are rich in bioactive ingredients with beneficial properties for preventing and managing diabetes [148].

Phenolic compounds are the predominant bioactive components in cherries and blueberries [40,149]. The difference in the total content of phenolics found between the various fruits relates to the cultivation, growth, maturity of the fruits, harvest, storage, and also the analytical method used for its quantification [150,151]. In the *Prunus* genus, namely in sweet cherry fruits, phenolic acids, flavonols, and anthocyanins are the main phenolics identified [40,152]. On the other hand, the most abundant phenolic compounds present in the fruits of the *Vaccinium* genus are procyanidins and phenolic acids; stilbene derivates (e.g., resveratrol), flavonols, and anthocyanins (e.g., malvidin, cyanidin, delphinidin, petunidin, and peonidin) [149,153]. In recent years, the phenolic compounds have demonstrated to be very promising agents in the control of hyperglycemia [146,154]. According to the literature data, flavonoids seem to be involved in the maintenance and function of pancreatic *β*-cells [145]. The reduction of oxidative stress, namely through increased endogenous antioxidant capacity, and the decreased reactive oxygen species (ROS) accumulation and translocation of pro-inflammatory cytokines, are among the main factors. Furthermore, the increase in anti-apoptotic gene expression, the reduction of caspase-3 and caspase-8, and protection against DNA damage have also been taken into account [145]. Different classes of phenolic compounds such as flavanol derivatives, flavan-3-ols, quercetins, anthocyanins, and phenolic acids have demonstrated the capacity to reduce ROS levels, inflammation, and protein glycation [155]. The inhibition of the major enzymes involved in carbohydrates’ metabolism, the increase in glucose transporters expression (GLUT) (e.g., GLUT-4), and the activation pathways responsible for signaling and secretion may be due to the action of these compounds [155]. Below are described the principal mechanisms, through which the phenolic compounds of cherries and blueberries can reduce hyperglycemia (Table 7).

### 6.1. Enzymes’ Inhibition

The inhibition of enzymes involved in the carbohydrates’ metabolism is one of the therapeutic strategies to control postprandial hyperglycemia, namely the *α*-glucosidase and *α*-amylase [143,171]. The reduction of hydrolysis and absorption of carbohydrates results in better blood glucose and the reduction of metabolic complications in diabetic patients [146].

Recent data reported that phenolic compounds can inhibit the activity of digestive enzymes [143,146]. Sweet cherry extracts from different cultivars were able to inhibit α-glucosidase in a dose-dependent manner (IC_50_ ranging from 10.25 ± 0.49 to 16.31 ± 0.71 µg/mL) [43]. Similarly, in a study with cherry by-products, it was demonstrated that the aqueous infusion of the stems was the most active against this enzyme (IC50 = 3.18 ± 0.23 μg/mL) [159]. In the in vitro study conducted by Ji et al. [156], it was observed that the anthocyanin-rich bilberry extract (BE) was able to inhibit the activity of α-glucosidase and α-amylase enzymes (IC_50_ = 0.31 ± 0.02 e 4.06 ± 0.12 mg/mL, respectively). Moreover, in the same study the hypoglycemic effect of BE extract on postprandial glucose was evaluated. The obtained results demonstrated that this extract was able to decrease hyperglycemia in the treated group with BE when compared to the control group [156]. In another work, fifteen different cultivars of *Vaccinium corymbosum* exhibited greater inhibitory activity against α-glucosidase with values ranging between 103.22 ± 0.15% and 184.70 ± 0.01% inhibition, and α-amylase ranging from 91.79 ± 0.82% and 103.32 ± 0.34% inhibition [157]. Moreover, a positive correlation between total phenolic content and inhibition of these enzymes was demonstrated [157]. Among anthocyanins present in cherries and blueberries, cyanidins were able to inhibit α-amylase, which controlled postprandial hyperglycemia [172].

The dipeptidyl-peptidase IV (DPP-IV) is the other enzyme involved in glucose homeostasis, which acts in the cleavage of peptide bonds and the release of compounds necessary for the formation of several molecules [143]. According to Jufeng et al. [160], resveratrol, luteolin, apigenin, and flavone are flavonoids present in blueberries with inhibitory effects on DPP-IV.

### 6.2. Pancreatic β-Cells Protection

Pancreatic *β*-cells are responsible for the production and release of insulin, the only hormone capable of lowering blood glucose concentration. The increase in oxidative stress can lead to damage in these cells, leading to impairment of their functions [143]. Hyperglycemia, dyslipidemia, inflammation, and autoimmunity are factors responsible for *β*-cells’ dysfunction and, consequently, lower insulin response [173]. When blood glucose levels are very high, there is an increase in ROS, glycation reactions, and apoptosis of pancreatic *β*-cells. In this sense, the search for bioactive compounds that can protect the cells, particularly phenolic compounds, has been a target of study by the scientific community [143,146,154].

Several studies have suggested that phenolic compounds can protect pancreatic *β*-cells mainly through their antioxidant potential [153,163,164]. Alloxan-induced diabetic rats were treated with ethanolic extracts of cherries (200 mg/kg) for 30 days, and it was observed that these extracts were able to reduce hyperglycemia and protect pancreatic β-cells from oxidative damage [162]. Similarly, in a study developed by Li et al. [163], C57BL/6J mice were supplemented with blueberry-leaf extract (1% *w*/*w* for 8 weeks) and the glucose homeostasis and insulin sensitivity were evaluated. The obtained results demonstrated that this extract was able to decrease glucose tolerance, body weight, plasma glucose, and glycated hemoglobin [163]. Moreover, it was found that the treated group increased the mRNA content of genes related to pancreatic *β*-cell proliferation, such as neurogenin 3 (Ngn3), V-maf musculoaponeurotic fibrosarcoma homolog A (MafA), paired box 4 (Pax4), and insulin 1 and 2 (Ins1-2). Pancreatic insulin signaling-related genes, namely the insulin receptor substrate (IRS), glucose transporter 2 (GLUT-2), phosphatidylinositol 4,5-bisphosphate 3-kinase catalytic subunit alpha isoform (PIK3ca), and phosphoinositide-dependent kinase (PKD1), were also increased. The transcriptional expression of genes related to the apoptosis of *β*-cells was diminished [163]. Resveratrol is one of the phenolics that can be found in blueberries, and it was described as able to protect against *β*-cell dysfunction [161]. According to Rouse et al. [161], this compound (0–10 µmol/L) limited the phosphodiesterase activity, increasing intracellular cAMP levels. Phosphodiesterase is responsible for cAMP degradation, and its inhibition prevents this degradation, allowing the insulin secretion and maintenance of pancreatic *β*-cells [161].

### 6.3. Insulin Release and Regulation

Insulin is synthesized by pancreatic *β*-cells and is involved in the regulation of glucose metabolism, through the glucose uptake in adipose tissue and skeletal muscles and the inhibition of gluconeogenesis in the liver [143]. Insulin secretion depends on the proper functioning of pancreatic *β*-cells. In this context, the use of natural products capable of promoting insulin secretion and reducing the adverse effects of synthetic drugs has been evaluated by the scientific community [174].

Anthocyanin, hydroxycinnamic, and flavonol-rich extracts (25 μg/mL) of sweet cherry promoted glucose consumption, insulin sensitivity, delayed glucose absorption, and inhibited excessive gluconeogenesis in HepG2 cells [165]. More recently, it was demonstrated that a blueberry-supplemented diet improved insulin sensitivity and glucose tolerance in high fat diet-induced obese mice [166]. In addition, the presence of small scattered islets in the blueberry-treated group was observed. This finding could mean that blueberries have the potential for the regeneration of pancreatic *β*-cells [166]. Similarly, previous preclinical studies demonstrated that dietary supplementation with blueberries in 32 obese, non-diabetic, and insulin-resistant subjects improved insulin sensitivity [167].

The expression of Sirtuin 1 (SIRT1) stimulates insulin secretion. According to Luo et al. [168], resveratrol (2.5, 12.5 µmol/L) improved insulin release from INS-1 cells by disrupting the SIRT1-uncoupling protein-2 (UCP2) axis. In another study, the effects of blueberry anthocyanin extract (5 or 10% by oral gavage: 10 mL/kg/day) were evaluated in streptozotocin (STZ)-induced diabetic rats, for 9 weeks [169]. SIRT1 expression, superoxide dismutase (SOD), and glutathione (GSH) activities were enhanced. In fermented berries beverages, anthocyanins (50 µmol cyanidin-3-glucoside equivalents), namely delphinidin-3-arabinoside, were responsible for the increased insulin secretion and the modulation of genes and proteins related to this action [170].

On the other hand, insulin resistance is one of the main consequences of diabetes, since high blood glucose levels reduce the ability of the cells to absorb and use the glucose for energy production. In addition to insulin, it is known that tyrosine kinase receptor is also involved in glucose homeostasis [143]. Tyrosine residues phosphorylation activates the insulin receptor substrate proteins (IRS1 and IRS2) and, consequently, regulates the insulin action [175]. Imbalance in this phosphorylation can lead to defects in the protein signaling of the IRS1 and IRS2 receptor substrates, leading to insulin resistance [175]. In this context, phenolic compounds demonstrated that they were able to improve insulin resistance.

### 6.4. Anti-Inflammatory Properties

Inflammation is a biological process of the immune system in response to infection or tissue injury. The expression of inflammatory cells, together with the production of ROS, provides a microenvironment advantageous to cell damage and apoptosis and, consequently, to the development of chronic diseases [176]. Thus, slowing down this process is essential; therefore, the phenolic compounds have been an alternative to consider [177].

In ancient times, cherry juice was used in the treatment of goats through the inhibition of inflammatory pathways [39]. In a study conducted by Kelley et al. [178], the effects of sweet cherries ingestion on plasma lipids and markers of inflammation in healthy humans was determined. After 28 days, a decrease in C-reactive protein (CRP), nitric oxide (NO), plasma uric acid, LDL, and TNF-*α* was verified. Similar results were obtained using Jerte Valley cherry-based beverage on the inflammatory load of rats and ringdoves, suggesting that this type of beverage up-regulated the levels of anti-inflammatory cytokines and down-regulated the levels of pro-inflammatory cytokines [179].

Recently, it was demonstrated that phenolic-targeted fractions of sweet cherry possess anti-inflammatory activity against RAW macrophages and AGS cells, capturing nitric oxide (NO), and decreasing NO synthase and cyclooxygenase-2-expression (COX-2) [180]. Similar results have already been obtained by Seeram et al. [181] when evaluating the effects of cyanidin alone and with other anthocyanins reported in cherries. The sweet cherries were able to inhibit COX-1 and COX-2 by an average of 28 and 47%, respectively [181]. Moreover, it was described that the inhibition of COX of anthocyanins of cherries and raspberries was analogous to those of some non-steroidal anti-inflammatory drugs (NSAIDs) at 10 µm concentration [176].

Regarding blueberries, several in vitro studies described the ability to modulate inflammatory markers in different types of cells [182,183,184]. Moreover, 48 patients with a metabolic syndrome who received a daily dose of 50 g of blueberry powder (equivalent to 350 g of fresh fruit) for 8 weeks demonstrated improvements in systolic and diastolic blood pressures [185]. However, the ingestion of 375 g of blueberries by athletic individuals 1 h prior to 2.5 h of treadmill running, for 6 weeks, demonstrated an increase in the anti-inflammatory regulatory cytokine IL-10 and a decrease in oxidative stress markers, following exercise [186].

Although cherries and blueberries present a good anti-inflammatory activity, the exact mechanisms that explain this potential still need to be explored.

## 7. Impact on Gut Microbiome

In recent decades, the relationship between food, health, and microbiota (mainly intestinal) has begun to be established, in which the consumption of a “Western diet” based on foods with a high content of rapidly assimilated sugars, a high quantity of animal proteins and a large proportion of fats produces an imbalance in intestinal populations, with an alteration of the microbial ecosystem at the population and functional level, resulting in health problems associated with early inflammatory and degenerative processes [187]. In the same way, that adherence to a Mediterranean diet, varied in vegetables, legumes, fruits, meat and fish, allows for maintaining a healthy microbiota with complete functionality associated with disease prevention, stimulation of the host’s immune system, neuromodulation, and improvement in the acquisition of nutrients [188]. These changes in the populations that make up the different microbiota are mainly associated with the nutritional profile or nutritional composition of the ingested foods that determine the dominance of one or another microbial group; therefore, those food constituent compounds that are used by the beneficial microbiota have been called prebiotics and their consumption is essential in the establishment and maintenance of a functional microbiota [189,190]. In this way, phenolic compounds are considered prebiotics. A total of 95% of them overcome the digestive processes intact and reach the intestine, even the colon, where they are biotransformed by beneficial species of the microbiome such as *Bifidobacterium* or *Akkermansia*. [191].

In this way, the consumption of foods rich in phenolic compounds seems to be a relevant factor in the dynamics of intestinal populations [192,193]. However, as indicated, polyphenol-rich beverages based on cherry (sweet cherry or tart cherry) and blueberry can have different presentations, with concentrations of bioactive compounds and simple preparations up to fermented preparations that can also affect the composition of phenolic compounds. In this way, studies related to the influence of the consumption of functional beverages made from these berries focus mainly on the use of juices or concentrates [194,195,196]. Most of the evidence about beverages rich in phenolic compounds focuses mainly on wine, grape juice, and beers rich in phenolic compounds, such as toasted beer [192,197], Although they have a different phenolic profile than what we can find in beverages made from cherry or blueberry, the greater knowledge about the influence of these beverages and their catabolism can help us understand the dynamics of the phenolic compounds that we find in beverages. For example, quercetin metabolized by *Eubacterium oxidoreducens* produces a decrease in the Firmicutes/Bacteroides ratio and the abundance of *Erysipelotrichaceae*, *Bacillus*, and *Eubacterium cylindroide* [198]. In a two-week study evaluating the consumption of a glass of beer (33 cl), it was observed that the consumption of non-alcoholic beer seems to have less influence on the intestinal microbiota, and that the consumption of black beer improves the proportion of *Akkermansia* related with polyphenols’ catabolism of this bacterial genus [192]. Although other studies analyzing the consumption of red wine demonstrate that its intake can affect the intestinal microbiome but that the metabolic capacities of individuals have a prominent importance in the modulation of populations, despite finding a homogenizing effect in the consumption of moderate amounts of this drink rich in phenolic compounds associated with these [193,197]. However, we must consider that the frequent consumption of alcoholic beverages, despite presenting phenolic compounds that may have positive implications for health, is related to the increase in the *Bacterioides* versus *Firmicutes* ratio, which is not always the appropriate trend for intestinal populations [198].

Thus, studies regarding the effect of the consumption of functional beverages made from cherry or blueberry on the intestinal microbiome are quite scarce because these beverages may not be as common as those mentioned, such as red wine and beer. For example, in a study conducted with adult men, evaluating the consumption of a drink rich in anthocyanins with characteristics that could be compared to a blueberry drink, consuming 750 mL/day for 55 days, a change in the microbiota was observed, as well as the functionality of it and a modification in the transcriptome of individuals towards the modulation of oxidative stress [195]. This study observed that the participants who consumed the beverage rich in anthocyanins increased the proportion of Bacterioides, and when performing an analysis of the functionality using PICRUSt, an enrichment was observed in the routes involved in the biosynthesis of phospholipids, metabolism of carbohydrates (starch, sucrose, frutose, and mannose) and amino acids (proline, arginine, lysine, and alanine). An increase in the number of pathways associated with DNA repair was also observed [195]. The consumption of powdered drinks based on blueberry, with a consumption of 25 g of the powdered drink in 250 mL for 6 weeks, generated an increase in the abundance of *Bifidobacterium* and Lactobacillus compared to the control group. This fact may be due to the fact that *Bifidobacterium* is a bacterial genus that is specialized in the catabolism of a variety of nondigestible plant polymers, glycoproteins, and glycoconjugates [199].

A study carried out on mice by applying Montmorency tart cherry juice, Balaton tart cherry juice, and sweet cherry juice for 21 days demonstrated substantial changes in the concentration of microorganisms, diversity indices, and specific variations associated with the concentration of the juice used. This study was carried out using five different concentrations of juice of each of the cultivars used (1/4, 1/7, 1/10, 1/15, 1/20 *v*/*v*). In the case of treatments with tart cherry, there is a decrease in the number of taxa as the concentration of the juice increases, unlike in the case of sweet cherry where there is a decrease to intermediate concentrations and then it increases again. Regarding the alpha diversity indices, this seems to behave in a similar way to the variation in the number of taxa. Analyzing the taxa detected individually in the three types of juices, a decrease in the concentration of Firmicutes and an increase in Verrucomicota was observed as the concentration of juice increased, associated with an increase in *Akkermansia* and *Barnesiella*, as well as a decrease in Bacteroides. It is also noteworthy that no correspondence was found between the variations in the concentration of Lactobacillus and the different treatments [196].

For example, some case studies analyze the use of functional beverages fermented with different microorganisms to improve their qualities; in this way, a synbiotic is achieved, which is a product that combines the properties of the probiotic and the prebiotic, and the beneficial properties for the health of the beverage with a high content of phenolic compounds [191,200]. These beverages are common in some regions such as China and stand out for a composition of probiotic bacteria and fungi such as *Lactobacillus*, *Issatchenkia,* and Saccharomyces, and the abundance of *Lactobacillus* and *Dekkera* was potentially related to the antioxidant activity of the traditional fermented blueberry beverages [201]. Although the content of anthocyanins and phenolic compounds are generally affected due to bacterial metabolism, a reduction in the former is observed, so a reduction in antioxidant capacity is expected [202]. In turn, this modification of the properties is associated with a reduction in the concentration of sugars, reducing the glycemic index [203]. For example, Cheng et al. [204] demonstrated that the use of a blueberry drink fermented with *Lactobacillus casei* was capable of improving intestinal functions and modifying the intestinal microbiota. In this case, an increase in the proportion of *Lactobacillus*, *Bifidobacterium*, *Ruminococcus*, *Akkermansia,* and the butyrate-producing bacteria in a dosage-dependent manner was observed, accompanied by an increase in the alpha diversity indices. The increase in bacterial species related to the synthesis of butyrate is especially relevant because there is a relationship between its presence and the reduction of inflammatory processes at the intestinal and systemic levels [205]. In addition, the increase in the proportion of *Akkermansia* and *Ruminococcus* is related to the improvement in the secretion of the mucus layer that is reduced by Western diets and is related to the maintenance of better intestinal function [204]. A positive interaction between the consumption of fermented blueberry beverages for 17 weeks and the modification of intestinal populations, with a reduction in the abundance of Firmicutes, *Oscillibacter*, and *Alistipes* (which are obesity-related bacteria) and an increase in *Akkermansia*, *Barnesiella*, *Olsenella*, *Bifidobacterium*, and *Lactobacillus* relative to the control based on a high-fat diet. In addition, the consumption of these fermented and unfermented beverages generated an increase in the alpha diversity of intestinal bacterial populations [206].

In this way, we can indicate that the consumption of functional beverages based on cherry and blueberry, with a high concentration of phenolic compounds, has a modulating effect on intestinal populations, increasing the abundance or proportion of beneficial genera such as *Akkermansia*, *Lactobacillus*, *Bifidobacterium*, or *Ruminococcus*, balancing the Firmicutes/Bacteroides ratio, and generating positive effects at the individual level derived from the intake of phenolic compounds but also associating with the improvement of the structure of the microbiome, acting as an efficient and easy-to-administer prebiotic (Figure 6).

## 8. Recent Advances and Future Perspectives and Opportunities for Functional Beverages

In the last decade, functional foods have demonstrated a marked growth. Foods and functional beverages formulated from natural products have excellent biological properties, as they are an interesting focus of research and development in the food and pharmaceutical industries [207]. This demand is due to many factors, such as the increased concern with health conditions, the awareness of the association between diet and health, and also busy and unhealthy lifestyles [208]. It is known that the consumption of fruits and vegetables helps in health promotion and prevention of several diseases, including diabetes, cardiovascular problems, and cancer [209]. A high standard of lifestyle in cosmopolitan cities and the concern to mitigate chronic diseases, request greater availability of functional foods to satisfy these needs [210].

The reduction of 20% in health-care expenses in the ingestion of functional food as one of the strategies for the public health improvement shows the economic potential of this type of food [211]. Phenolic compounds, minerals, fibers, vitamins, and probiotic bacterial strains are some of the components that make a beverage “functional”. New methods for beverage enrichment with novel constituents through “omics” technologies have been explored [212]. In addition, the continuous development of the functional beverages market depends on these products’ present health claims [210]. A health claim can be defined as a description of what the product does in terms of health, well-being, and performance [213]. The beneficial health effects and detailed information about beverages’ formulation will influence consumer knowledge and attitude. Moreover, the health claims influence the purchasing decisions of consumers and help in making more informed choices [214].

Among the different categories of functional foods, beverages are one of the most active. Easy delivery and storage, a greater possibility of incorporating desirable bioactive compounds and nutrients, and the ability to respond to the needs of consumers through appearance, size, shape, and content are key factors for this growing development [210]. The functional beverages’ industry depends on consumers’ recognition of the linkage between health and diet. Nowadays, the baby-boom generation that is ageing is willing to buy and consume products that improve their health. In this context, immunity enhancement has become the main focus of functional beverages producers and consumers [210]. The addition of bioactive compounds and other functional components, and/or reduction of the other constituents, such as fat or sugar, are part of the functional foods’ formulations [210]. Thus, the production and sale of beverages supplemented with probiotics, natural antioxidants, and red fruits such as cherries and blueberries have been increased [215].

The increase in consumer interest in organic constituents instead of artificial components is another advantageous factor for the functional beverage development market [216]. This behavior allows the industry to enhance its products through supplementation with natural products, namely those based on red fruits, with proven health benefits. Regarding the positive factors that drive the development of functional drinks, the following should be highlighted: consumer age, education, decrease in health-care costs, access to information, media, nutrition labeling, aroma and texture, mood, beliefs, and appetite [210]. The adoption and maintenance of a healthy lifestyle have led to an increase in the demand for functional drinks and, consequently, to the development of new products that meet the needs of today’s society, satisfying several criteria that make marketing more efficient [211]. In addition, the increase in life expectancy increases the need for a better quality of life with healthy food choices [211]. Functional beverages such as energetic juices, supplemented waters, diet drinks, and antioxidant juices are opportunities for the growth of this market type.

One of the great future challenges in the functional food market is the lack of an internationally accepted definition of the functional food and beverage [210]. However, despite this difficulty, the global value of the functional foods market has been progressively increasing with an average growth rate of 8.5%. U.S. and Japan are the main markets, followed by Europe [217]. Functional beverages are the fastest-growing sector within functional foods. According to Bagchi and Nair, 2016 [218], it is estimated that by 2025, functional beverages will account for 40% of the overall consumer demand. The socio-cultural and economic differences of consumers also influence the growth of these beverages’ market [210].

The development of new functional beverages and the improvement of existing ones are other interesting focuses of research and innovation. In recent years, the potential of probiotics, prebiotics, natural compounds, and by-products of fruits in the production of new beverages without affecting their functionality has been studied [37]. Fruits-based beverages, vegetables, cereals, and grapes are examples of functional beverages with several bioactive compounds [210]. However, it is important to evaluate the interactions between the various ingredients in the same product. These interactions can result in insolubility, oxidation, precipitation, degradation, and loss of functionality of the beverage [211]. The metabolism and bioavailability must be carefully evaluated [219].

Another challenge in the research and development of functional beverages is the knowledge of the ideal dosage of bioactive or functional constituent to be added. The main goal is for it to be a sufficient dosage to produce a beneficial effect without negatively affecting other components of the formulation [210]. On the other hand, it is also essential to ensure that the compounds will remain intact, active, and bioavailable after processing and storage [220]. In an attempt to improve functional beverages, it is necessary to improve technology in the production process, such as microencapsulation [208,221] and metabolomics [222].

Functional beverages with antimicrobial and antioxidant compounds incorporated are an area to be explored. Bacteriocins, oligosaccharides, and polysaccharides are natural antimicrobials able to substitute chemical preservatives [223]. These compounds, in addition to preservative capacity, also possess health-promoting properties [224]. Furthermore, the phenolic compounds also have excellent health-promoting properties [225].

Finally, the use and valorization of fruit and vegetable by-products have been studied by several authors [146,226,227]. These bioresidues are very rich in phenolic compounds, minerals, vitamins, and organic compounds, among others with proven beneficial properties [228]. These compounds can be explored, as they demonstrate promising potential in the area of food and functional beverages.

In sum, the development of new functional beverages with new constituents/ingredients and functionalities can have very promising results for the health of consumers. Furthermore, the demand for products with health-promoting properties will continue to increase. Despite the challenges and difficulties inherent to beverages and functional food, the help of new technology and the progress of science in the exploration of new bioactive compounds are essential for the advancement of this market niche.

## 9. Conclusions

Functional foods rich in phenolics, including their derivative beverages, seem to be a promising strategy and an added value in current days. Among fruits, cherries and blueberries have been a target of many studies, not only owing to their organoleptic properties, such as color, aroma, and taste, but also due to their effectiveness in reducing the risk of occurrence of many disorders, attenuating their symptoms and/or retarding their development. These effects are closely attributed to the ability of their phenolics, standing out anthocyanins, to counteract oxidative stress and interact with inflammatory pathways, and in this way, promote a healthy status. Functional beverages possess several advantages, such as lower price and facility of ingestion. However, more detailed studies, including clinical trials and stability experiments, need to be conducted to increase their bioacessibility and reveal the safe and ideal dosage.

## Figures and Tables

**Figure 1 molecules-27-03294-f001:**
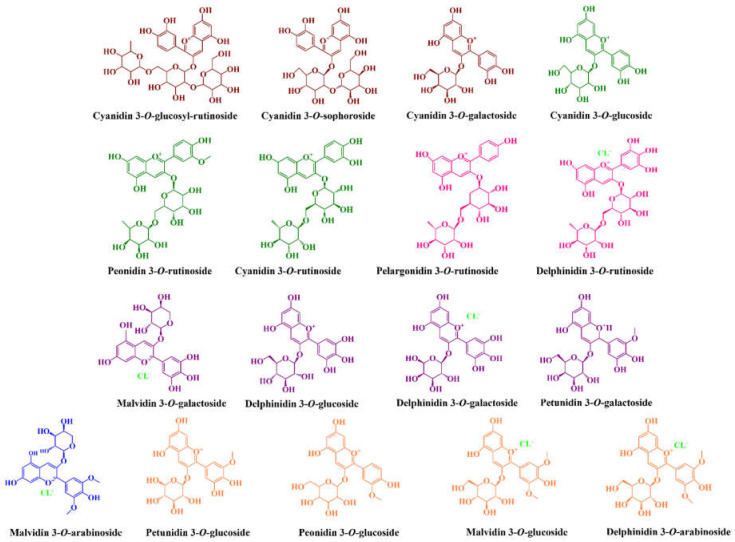
Main anthocyanins reported in tart cherries (brown color), sweet cherries (pink color), and both cherries (green color), highbush blueberries (purple color), rabbiteye blueberries (blue color), and both blueberries (orange color). (Figure created with ChemDraw Professional 16.0 (CambridgeSoft, Perkin Elmer Inc., Waltham, MA, USA)).

**Figure 2 molecules-27-03294-f002:**
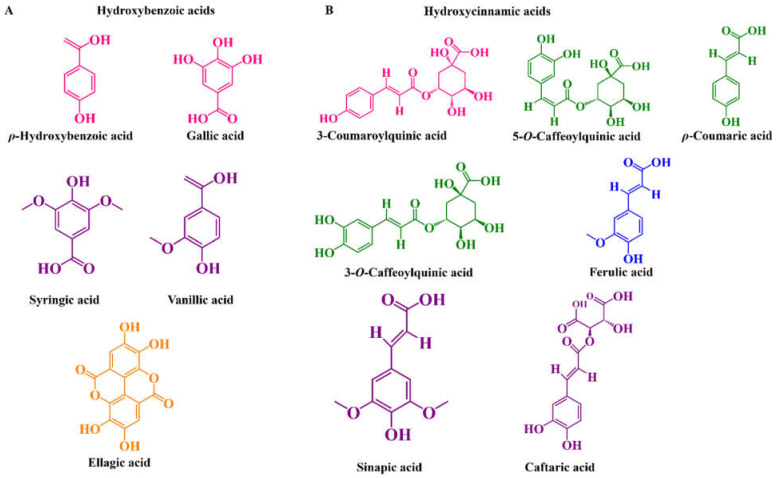
Main hydroxybenzoic (**A**) and hydroxycinnamic (**B**) acids reported in sweet cherries (pink color), and both tart and sweet cherries (green color), highbush blueberries (purple color), rabbiteye blueberries (blue color), and both blueberries (orange color). Vanillic, caftaric and sinapic acids were only detected in highbush blueberries [60,74,91]. (Figure created with ChemDraw Professional 16.0 (CambridgeSoft, Perkin Elmer Inc., Waltham, MA, USA)).

**Figure 3 molecules-27-03294-f003:**
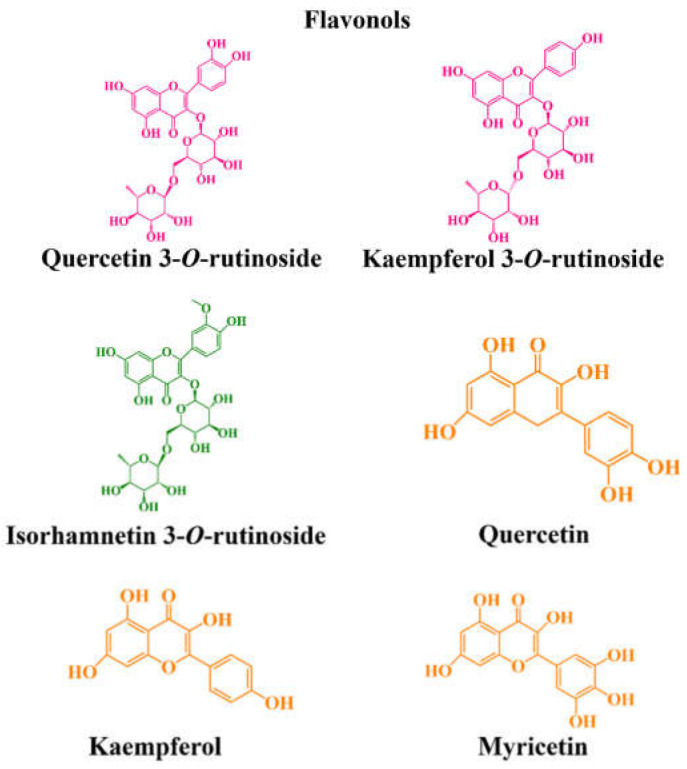
Main flavonols reported in sweet cherries (pink color) and both tart and sweet cherries (green color), and both highbush and rabbiteye blueberries (orange color). (Figure created with ChemDraw Professional 16.0 (CambridgeSoft, Perkin Elmer Inc., Waltham, MA, USA)).

**Figure 4 molecules-27-03294-f004:**
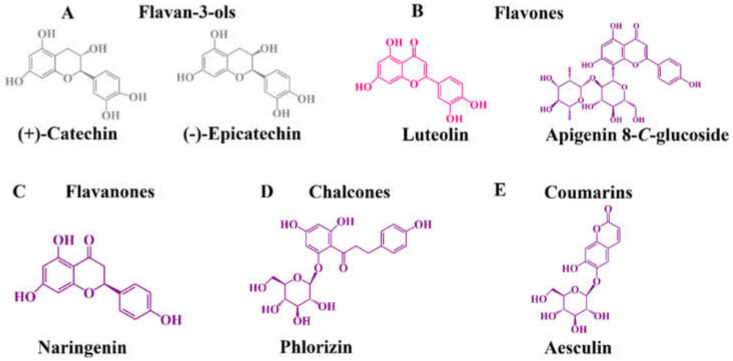
Main flavan-3-ols (**A**), flavones (**B**), flavanones (**C**), chalcones (**D**), and coumarins (**E**) reported in both cherries and blueberries (grey color), sweet cherries (pink color), and highbush blueberries (purple color). (Figure created with ChemDraw Professional 16.0 (CambridgeSoft, Perkin Elmer Inc., Waltham, MA, USA)).

**Figure 5 molecules-27-03294-f005:**
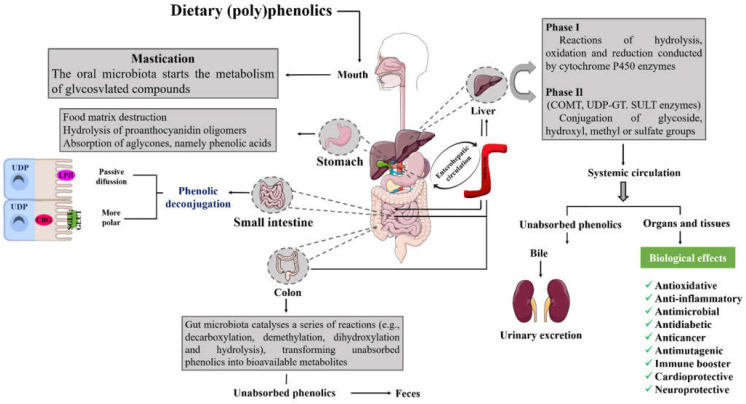
Schematic representation concerning phenolics’ absorption, metabolism, distribution, and excretion (COMT, catechol O-metiltransferase; UDP, uridine 5′-diphospho-glucuronosyltransferase; UDP-GT, glucuronil-transferase; SULT, sulfotransferase; LPH, lactase-phlorizin hydrolase; SGLT, sodium-dependent glucose cotransporters, GLUT, glucose transporters; CBG, cytosolic β-glucosidase). Figure created with Smart Servier Medical Art tools (https://smart.servier.com, 1 January 2022).

**Figure 6 molecules-27-03294-f006:**
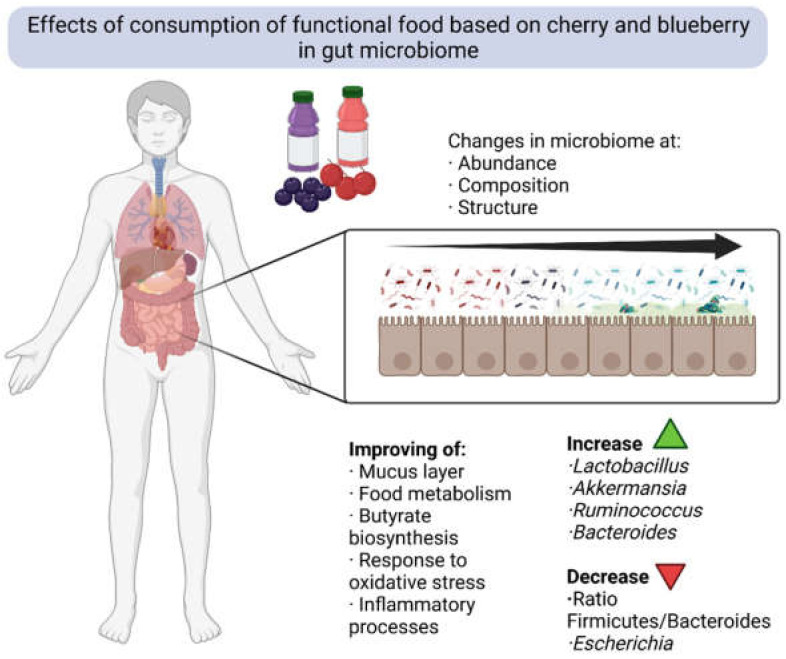
Main effects of functional beverages based on cherry and blueberry in dynamics and functionality of gut microbiome.

**Table 1 molecules-27-03294-t001:** The Japanese FOSHU criteria for functional food.

They are food (not capsules, pills, or powder) based on naturally occurring food components They can and should be consumed as part of the normal daily diet They have a defined function on the human organism: To improve immune function To prevent specific diseases To support recovery from specific diseases To control physical and physic complaints To slow down the ageing process

**Table 2 molecules-27-03294-t002:** The FUFOSE definition of functional food in Europe [13].

Functional foods are:
Conventional or everyday food consumed as part of the normal diet Composed of naturally occurring components, sometimes in increased concentration or present in foods that would not normally supply them Scientifically demonstrated to promote positive effects on target functions beyond basic nutrition Thought to provide enhancement of the state of well-being and health in order to improve the quality of life and = or reduce the risk of disease Advertised by authorized claims

**Table 5 molecules-27-03294-t005:** Categories of functional foods.

Category	Example
Basic food	Carrots (containing the antioxidant *β*-carotene); Turmeric (containing curcumin); Grapes (containing resveratrol)
Processed foods	Oat bran cereal
Processed foods with added ingredients	Calcium-enriched fruit juice; margarine enriched in phytosterols; Beverages enriched with vitamins and minerals
Food enhanced to have more of a functional component	Tomatoes with a higher level of lycopene
Isolated, purified preparations of active food ingredients (dosage form)	Isoflavones from soy *β*-Glucan from oat bran Anthocyanins from red fruits

**Table 6 molecules-27-03294-t006:** Nutritional composition and main phenolic compounds found in tart and sweet cherries, and highbush and rabbiteye blueberries (mg per 100 g of fresh weight (fw)) and juices (mg/L).

	Fruits				Juices			
	Tart Cherries	Sweet Cherries	Blueberries		Tart Cherries	Sweet Cherries	Blueberries	References
**Basic chemical composition**								
Water (g per 100 g)	86.1	82.2	84.2		85.2	85.0	89.7	[86]
Energy (kcal per 100 g)	50.0	63.0	57.0		59.0	54.0	37.0	
Energy (kJ per 100 g)	209.0	263.0	240.0		248.0	226.0	-	
**Macronutrients**								
Total protein (g per 100 g)	1.0	1.1	0.74		0.31	0.91	0.48	[86]
Betaine (mg per 100 g)	-	-	0.20		-	-	-	
Tryptophan (mg per 100 g)	-	0.009	0.030		-	-	-	
Threonine (mg per 100 g)	-	0.022	0.020		-	-	-	
Isoleucine (mg per 100 g)	-	0.020	0.023		-	-	-	
Leucine (mg per 100 g)	-	0.030	0.044		-	-	-	
Lysine (mg per 100 g)	-	0.032	0.013		-	-	-	
Methionine (mg per 100 g)	-	0.010	0.012		-	-	-	
Cystine (mg per 100 g)	-	0.010	0.008		-	-	-	
Phenylalanine (mg per 100 g)	-	0.024	0.026		-	-	-	
Tyrosine (mg per 100 g)	-	0.014	0.009		-	-	-	
Valine (mg per 100 g)	-	0.024	0.031		-	-	-	
Arginine (mg per 100 g)	-	0.018	0.037		-	-	-	
Histidine (mg per 100 g)	-	0.015	0.011		-	-	-	
Alanine (mg per 100 g)	-	0.026	0.031		-	-	-	
Aspartic acid (mg per 100 g)	-	0.569	0.057		-	-	-	
Glutamic acid (mg per 100 g)	-	0.083	0.091		-	-	-	
Glycine (mg per 100 g)	-	0.023	0.031		-	-	-	
Proline (mg per 100 g)	-	0.039	0.028		-	-	-	
Serine (mg per 100 g)	-	0.03	0.022		-	-	-	
Total lipids (g per 100 g)	0.3	0.2	0.33		0.54	0.02	0.21	
Fatty acids, total saturated (g per 100 g)	0.068	0.038	0.028		-	0.004	0.018	
SFA 14:0 (g per 100 g)	0.002	0.001	-		-	-	-	
SFA 16:0 (g per 100 g)	0.048	0.027	0.017		-	0.003	0.011	
SFA 18:0 (g per 100 g)	0.016	0.009	0.005		-	0.001	0.003	
Fatty acids, total monounsaturated (g per 100 g)	0.082	0.047	0.047		-	0.005	0.031	
MUFA 16:1 (g per 100 g)	0.001	0.001	0.002		-	-	0.001	
MUFA 18:1 (g per 100 g)	0.081	0.047	0.047		-	0.005	0.031	
Fatty acids, total polyunsaturated (g per 100 g)	0.09	0.052	0.146		-	0.006	0.095	
PUFA 18:2 (g per 100 g)	0.046	0.027	0.088		-	0.003	0.057	
PUFA 18:3 (g per 100 g)	0.044	0.026	0.058		-	0.003	0.038	
Carbohydrates (g per 100 g) (by difference)	12.2	16	14.5		13.7	13.8	9.42	
Total ash (g per 100 g)	0.4	0.48	0.24		0.28	0.031	-	
Total dietary fiber (g per 100 g)	1.6	2.1	2.4		-	1.5	1.6	
Total sugars (g per 100 g)	8.49	12.8	9.96		12.2	12.3	6.47	
Fructose (g per 100 g)	3.51	5.37	4.97		4.95	-	-	
Glucose (g per 100 g)	4.18	6.59	4.88		7.26	-	-	
Sucrose (g per 100 g)	0.8	0.15	0.11		-	-	-	
Lactose (g per 100 g)	-	-	-		-	-	-	
Maltose (g per 100 g)	-	0.12	-		-	-	-	
Galactose (g per 100 g)	-	0.59	-		-	-	-	
Starch (g per 100 g)	-	0	0.03		-	-	-	
**Micronutrients**								
**Minerals**								
Calcium (mg per 100 g)	16.0	13.0	6.0		13.0	14.0	5.0	[86]
Iron (mg per 100 g)	0.32	0.36	0.28		0.42	0.58	0.18	
Magnesium (mg per 100 g)	9.0	11.0	6.0		11.0	12.0	4.0	
Phosphorus (mg per 100 g)	15.0	21.0	12.0		17.0	22.0	8.0	
Potassium (mg per 100 g)	173.0	222.0	77.0		161	131.0	50.0	
Sodium (mg per 100 g)	3.0	-	1.0		4.0	3.0	2.0	
Zinc (mg per 100 g)	0.1	0.07	0.16		0.03	0.1	0.1	
Cooper (mg per 100 g)	0.104	0.06	0.057		0.042	0.073	0.04	
Manganese (mg per 100 g)	0.112	0.07	0.336		0.06	0.061		
Fluoride (μg per 100 g)	-	2.0	-		-	-	-	
Selenium (μg per 100 g)	-	-	0.1		-	0	0.1	
**Vitamins**								[86]
Vitamin C (mg per 100 g)	10.0	7.0	9.7		-	2.5	6.3	
Thiamin (mg per 100 g)	0.03	0.027	0.037		0.06	0.018	0.024	
Riboflavin (mg per 100 g)	0.04	0.033	0.041		-	0.024	0.027	
Niacin (mg per 100 g)	0.4	0.154	0.418		-	0.406	0.272	
Pantothenic acid (mg per 100 g)	0.143	0.199	0.124		-	0.127		
Vitamin B6 (mg per 100 g)	0.044	0.049	0.052		0.037	0.03	0.034	
Folate, total (μg per 100 g)	8.0	4.0	6.0		-	4.0	4.0	
Folate, DFE (μg per 100 g)	8.0	4.0	6.0		-	4.0	4.0	
Folate, food (μg per 100 g)	8.0	4.0	6.0		-	4.0	4.0	
Choline (mg per 100 g)	6.1	6.1	6.0		-	4.7	3.9	
Vitamin A, RAE (μg per 100 g)	64.0	3.0	3.0		-	6.0	2.0	
Vitamin A, IU (IU per 100 g)	1280.0	64.0	54.0		-	125	-	
Vitamin D (D2 + D3) IU (IU per 100 g)	-	64.0	-		-	-	-	
*β*-Carotene (μg per 100 g)	770.0	38.0	32.0		-	75.0	21.0	
Lutein + zeaxanthin (μg per 100 g)	85.0	85.0	80.0		-	57.0	52.0	
Vitamin E (mg per 100 g)	0.07	0.07	0.57		-	0.23	-	
*β*-Tocopherol (mg per 100 g)	-	0.01	0.01		-	-	-	
γ-Tocopherol (mg per 100 g)	-	0.04	0.36		-	-	-	
Δ-Tocopherol (mg per 100 g)	-	-	0.03		-	-	-	
γ-Tocotrienol (mg per 100 g)	-	0.04	0.07		-	-	-	
Vitamin K (μg per 100 g)	2.1	2.1	19.3		-	1.4	12.5	
**Phenolic Profile**								
	**Cherries**		**Blueberries**		**Juices**			
	**Tart Cherries**	**Sweet Cherries**	**Highbush**	**Rabbiteye**	**Tart Cherries**	**Sweet Cherries**	**Blueberries**	
TPC (mg GAE per 100 g fw)	275.3–652.27	28.3–493.6	2.7–585.3	390.0–2625.0	1510.0–2550.0 ^a^	582.7–4757.9 ^a^	1.65	[53,55,56,57,58,59,60,61,62,63,64,65,66,87]
TAC (mg C3G per 100 g fw)	15.5–295.0	3.7–98.4	34.5–552.2	69.97–378.31	553.0 ^a^	85.1–1095.9 ^a^	29.00–32.73 ^a^	[53,54,64,68,69,70,71]
**Anthocyanins**								
Cyanidin 3-*O*-glucosyl-rutinoside	89.0–227.66	-	-	-	92.86–441.11	-	-	[54,72,73,74,87,88]
Cyanidin 3-*O*-rutinoside	1.76–74.7	0.20–389.9	-	-	0.38–85.5	104.0–210.0	-	[40,43,54,72,73,76,77,87,88,89,90]
Cyanidin 3-*O*-sophoroside	0.13–10.44	t.r.	-	-	1.62–292.21	-	-	[54,72,73,74,76,87,88]
Cyanidin 3-*O*-glucoside	0.01–142.03	0.0–142.03	0.11–3.09	t.r.–8.20 ^c^	2.0–9.9	22.0–37.0	0.26–89.0	[40,43,54,59,66,72,73,76,77,81,84,85,87,89,90,91]
Cyanidin 3-*O*-xylosylrutinoside	t.r.	-	-	-	-	-	-	[53]
Cyanidin 3-coumaroyl-diglucoside	-	0.001–0.44	-	-	-	-	-	[76]
Cyanidin 3-5-diglucoside	-	0.16–1.05	-	-	-	-	0.0–28.5	[76,92]
Cyanidin 3-*O*-hexoside	-	-	19.23	-	-	-	-	[82]
Cyanidin 3-*O*-galactoside	0.0–2.63	t.r.	0.80–9.96	5.40–8.90 ^c^	-	-	13–59	[59,74,75,81,84,85,91]
cyanidin 3-(6″-acetyl-glucoside)	-	-	-	-	-	-	20.0	[93]
Cyanidin 3-*O*-sambubioside	-	0.09–0.16	-	-	-	-	-	[76]
Cyanidin 3-*O*-arabinoside	-	0.25–0.40	0.42–1.09	2.62 ^c^	-	-	1.40–160	[66,76,81,85,91]
Petunidin 3-*O*-galactoside	-	-	2.57–28.54	6.94 ^c^	-	-	0.34–125	[59,66,81,85,91]
Petunidin 3-*O*-glucoside	-	-	0.67–25.14	t.r.–9.93 ^c^	-	-	7.70–365.0	[59,81,82,84,85,91,93]
Petunidin 3-*O*-arabinoside	-	-	1.82–12.70	3.5.–4.30 ^c^	-	-	0.53–59.0	[59,66,81,84,91]
Petunidin 3-(6″-acetyl)glucoside	-	-	-	-	-	-	57.0	[93]
Peonidin 3-*O*-glucoside	t.r.	0.0–0.38	12.00–54.37	17.6–30.3 ^c^	-	-	0.63–91.0	[59,66,76,78,84,90,91,93]
Peonidin 3-*O*-rutinoside	-	0.0–6.7	-	-	-	29.0–36.0	-	[54,78,89,90]
Peonidin 3-*O*-pentose	-	-	0.52–0.69	-	-	-	-	[81]
Peonidin 3-*O*-galactoside	-	-	0.77–125.79	2.90–3.80 ^c^	-	-	0.54–19.0	[59,66,81,84,85,91]
Peonidin 3-*O*-arabinoside	-	-	-	2.4–13.4 ^c^	-	-	<1–2	[84,85,91]
Peonidin 3-(6″-acetyl)galactoside	-	-	-	-	-	-	6.0	[93]
Peonidin 3-(6″ -acetyl)glucoside	-	-	-	-	-	-	40.0	[93]
Pelargonidin 3-*O*-rutinoside	-	0.0–7.97	-	-	0.11–131.42	7.0–9.0	-	[76,78,88,89,90]
Pelargonidin 3-*O*-glucoside	-	0.22–0.71	-	-	-	-	10.1–35.6	[76,92]
Malvidin 3-*O*-glucoside	-	0.08–0.45	0.68–34.75	21.53 ^c^	-	-	6.25–271.0	[59,66,78,81,85,91,93]
Malvidin 3-*O*-galactoside	-	-	12.11–67.45	19.57 ^c^	-	-	6.0–160	[59,66,80,81,85,91,93]
Malvidin 3-*O*-arabinoside	-	-	6.77–9.41	4.64–17.80 ^c^	-	-	4.60–73.0	[66,81,84,85,91,93]
Malvidin-3-(6″-acetyl-galactoside)	-	-	0.99–1.74	-	-	-	34.0	[81,93]
Malvidin 3-*O*-xyloside	-	-	0.56	-	-	-	-	[81]
Malvidin-3-(6″-acetyl) glucoside	-	-	1.63	-	-	-	131.0	[81,93]
Malvidin 3-*O*-glucoside-acetaldehyde	-	0.08–0.11	-	-	-	-	-	[76]
Delphinidin 3-*O*-rutinoside	t.r.	t.r.	-	-	-	-	-	[53]
Delphinidin 3-*O*-glucoside	-	-	1.21–53.62	0.2–8.08 ^c^	-	-	7.70–365.0	[59,81,84,85,91,92]
Delphinidin 3-*O*-galactoside	-	-	2.29–53.29	7.97–16.3 ^c^	-	-	0.14–223.0	[59,66,81,84,85,91,93]
Delphinidin 3-*O*-arabinoside	-	-	1.66–41.07	4.67–5.6 ^c^	-	-	0.67–134.0	[59,66,81,84,91]
Delphinidin 3-(6″-acetyl)glucoside	-	-	-	-	-	-	2.0	[93]
Delphinidin 3-(malonyl)glucoside	-	-	-	-	-	-	86.0	[93]
Delphinidin	0.01–0.52	-	8.51–141.1	-	-	-	5.40–25.7	[70,72,82,83,92]
Malvidin	0.27–8.31	0.04–0.06	131.3–154.6	-	-	-	0.37	[66,70,72,79,83]
Peonidin	0.01–0.19	0.11–3.93	14.28–36.9	-	-	-	-	[70,72,79,82,83]
Cyanidin	3.41–6.64	0.04–0.18	21.17–66.3	-	-	-	0.09	[66,70,72,79,82,83]
Pelargonidin	1.35–64.36	-	-	-	-	-	-	[70,72]
Petunidin	-	-	1.78–87.6	-	-	-	-	[82,83]
**Hydroxybenzoic acids**								
*ρ*-Hydroxybenzoic acid	-	10.3–19.1	0.054–59.89	0.0–103.67	-	-	t.r.	[63,64,77,90,94,95]
Protocatechuic acid	-	0.054–3.28	5.22–41.45	-	-	-	-	[63,76,77,96]
Hydroxybenzoic acid-glycoside	-	0.15–0.32	-	-	-	-	-	[76]
Hydroxybenzoyl hexose	-	0.14–0.70	-	-	-	-	-	[76]
Vanillic acid-glycoside	-	0.76–3.05	-	-	-	-	-	[76]
Vanillic acid	-	-	0.011–0.027	-	-	-	t.r.	[94,95]
Syringic acid	-	0.0–0.071	0.034–9.95	-	-	6.64–14.46	t.r.	[63,77,94,95,96,97]
Gallic acid	-	0.0018–10.64	0.02–5.68	1.53–258.9	-	0.0–6.55	-	[60,63,64,77,78,96,97,98]
Ethyl gallate	-	0.0003–0.0014	-	-	-	-	-	[96]
Propyl gallate	-	0.0005–0.0099	-	-	-	-	-	[96]
Ellagic acid	-	-	0.75–6.65	0.0–19.25	-	-	-	[60,64]
2,5-Dihydroxybenzoic acid	-	0.0–1.50	-	-	-	-	-	[78,96]
**Hydroxycinnamic acids**								
Salicylic acid	-	0.0037–1.31	-	-	-	-	-	[90,96]
Cinnamic acid	-	7.8–11.1	0.003–0.07	-	-	-	-	[90,94]
Ferulic acid	-	0–5.7	0.018–4.16	0.0–16.97	1.14–1.27	1.01–6.35	t.r.	[60,64,77,90,94,95,96,97,99]
3-Caffeoylquinic acid	5.24–27.79	38.0–187.0	0.46–7.12	0.039–2.46	82.0–183.0	24.77–37.78	-	[54,60,74,76,77,81,90,97,100]
4-Caffeoylquinic acid	-	2.6–29.2	-	-	-	-	-	[90]
5-Caffeoylquinic acid	0.58–60.33	0.21–120.8	13.52–65.24	-	28.30–995	-	t.r.	[54,63,74,76,90,95,99,100,101]
3-Coumaroylquinic acid	-	37.0–452.52	-	-	91.0–555.0	-	-	[76,90,100]
4-Coumaroylquinic acid *cis*	-	0.74–18.58	-	-	-	-	-	[76]
4-Coumaroylquinic acid *trans*	-	4.92–19.46	-	-	-	-	-	[76]
5-Coumaroylquinic acid *cis*	-	0.38–0.96	-	-	12.0–81.0	-	-	[76,100]
5-Coumaroylquinic acid *trans*	-	0.53–1.53	-	-	-	-	-	[76]
3-Feruloylquinic acid *cis*	-	0.64–2.30	-	-	-	-	-	[76]
3-Feruloylquinic acid *trans*	-	0.72–5.86	-	-	-	-	-	[76]
4-Feruloylquinic acid *cis*	-	0.18–0.49	-	-	-	-	-	[76]
5-Feruloylquinic acid *trans*	-	0.04–0.25	-	-	-	-	-	[76]
5-Feruloylquinic acid *cis*	-	0.11–2.92	-	-	-	-	-	[76]
Caffeoylquinic acid glycoside	-	0.11–1.71	-	-	-	-	-	[76]
3,5-diCaffeoylquinic acid	-	0.26–2.87	-	-	-	-	-	[76]
4,5-diCaffeoylquinic acid	-	0.09–0.78	-	-	-	-	-	[76]
Caffeoylshikimic acid	-	0.26–0.56	-	-	-	-	-	[76]
3- and 4-Caffeoylquinic lactone	-	0.39–2.26	-	-	-	-	-	[76]
Caftaric acid	-	-	4.71	-	-	-	-	[77]
3-Coumaroylquinic lactone	-	0.39–0.99	-	-	-	-	-	[76]
4-Coumaroylquinic lactone	-	0.11–2.02	-	-	-	-	-	[76]
3-Coumaroyl-5-caffeoylquinicacid	-	0.03–0.76	-	-	-	-	-	[76]
3-Caffeoyl-4-coumaroylquinic acid	-	0.02–0.46	-	-	-	-	-	[76]
Coumaroyl hexose	-	2.95	-	-	-	-	-	[76]
Caffeoyl hexose	-	0.32–2.02	0.19–0.22	-	-	-	-	[76,81]
Caffeic acid	-	0.0–0.83	0.042–32.3	-	5.39–15.50	3.74–4.00	t.r.	[64,77,94,95,96,97,99]
Caffeic acid glycoside	-	0.52–8.79	-	-	-	-	-	[76]
Caffeoyl alcohol 3/4-*O*-hexoside	-	0.7–0.78	-	-	-	-	-	[76]
*ρ*-Coumaric acid	0.89–50.69	0.11–70.45	2.40–25.49	0.0–15.78	11.30–12.10	0.0–0.15	-	[54,60,63,64,74,77,78,96,97,99]
Feruloyl hexose	-	0.33–0.39	0.91–1.63	-	-	-	-	[76,81]
Sinapoyl hexose	-	0.20–0.50	-	-	-	-	-	[76]
Sinapic acid	-	-	0.005–0.11	-	-	-	-	[63,94]
3-*O*-Coumaroyl quinic acid	-	0.35–1.7	-	-	-	-	-	[102]
3-*O*-Coumaroyl quinic acid II	-	3.1–15	-	-	-	-	-	[102]
5-*O*-Feruloyl quinic acid	-	0.34	-	-	-	-	-	[102]
3-*O*-Feruloyl quinic acid	-	0.038–0.44	-	-	-	-	-	[102]
Chlorogenic acid isomer II	-	0.24–0.63	-	-	-	-	-	[102]
4-*O*-Coumaroyl quinic acid I	-	0.12–0.61	-	-	-	-	-	[102]
4-*O*-Coumaroyl quinic acid II	-	0.50–3.0	-	-	-	-	-	[102]
Malonyl-dicaffeoylquinic acid	-	-	0.76	-	-	-	-	[81]
Malonyl-caffeoylquinic acid	-	-	9.32	-	-	-	-	[81]
**Flavonols**								
Myricetin	-	0.0005–0.014	6.72–6.98	0.0–8.62	-	-	-	[64,96]
Myricetin 3-*O*-glucuronide	-	-	-	91.0–482.0 ^b^	-	-	-	[51]
Myricetin 3-*O*-galactoside	-	-	-	44.0–564.0 ^b^	-	-	-	[51]
Myricetin-3-(6″-rhamnosyl)galactoside	-	-	-	0.0–1.1 ^b^	-	-	-	[51]
Myricetin 3-*O*-glucoside	-	-	-	66.0–121.0 ^b^	-	-	-	[51]
Myricetin-3-(6″-rhamnosyl)glucoside	-	-	-	0.0–210.0 ^b^	-	-	-	[51]
Isomers of myricetin 3-*O*-pentoside	-	-	-	0.0–110.0 ^b^	-	-	-	[51]
Myricetin 3-*O*-rhamnoside	-	-	-	0.0–971.0 ^b^	-	-	-	[51]
Quercetin	-	0.0–2.51	0.29–21.48	0.046–9.97	184.0–739.0 ^b^	-	51.2	[51,60,64,77,90,94,96,103]
Quercetin 3-*O*-galactoside	-	-	0.19–31	269.0–1174.0 ^b^	0.0–4.0	-	-	[51,77,94,100]
Quercetin 3-*O*-glucuronide	-	-	0.06–1.76	475.0–3353.0 ^b^	-	-	-	[51,63,77]
Quercetin 3-*O*-hexoside	-	0.99–1.39	-	-	-	-	-	[76]
Quercetin 3-*O*-arabinoside	-	-	0.58	-	0.0–16.0	-	-	[81,100]
Quercetin 3-*O*-glucoside	0.21–0.44	0.0–26.55	0.9–34.64	81.0–203.0 ^b^	-	-	t.r.	[51,54,63,76,77,81,95]
Quercetin-3-*O*-[4″-(3-hydroxy-3-methylglutaroyl)]-α-rhamnoside syringetin-3-rhamnoside	-	-	-	0.0–1719.0 ^b^	-	-	-	[51]
Isomers of quercetin 3-*O*-pentoside	-	-	-	0.0–1005.0 ^b^	-	-	-	[51]
Quercetin 3-*O*-rhamnoside	-	-	26.0	88.10–4292.0 ^b^	18.0–45.0	-	-	[63,100]
Quercetin 3-*O*-rutinoside	0.84–7.63	0.78–51.97	0.008–0.056	0.044–6.74	4.10–53.80	0.0–4.74	65.0	[54,60,74,76,94,97,99,102,103]
Quercetin 7-*O*-glucoside-3-*O*-rutinoside	-	0.08–5.56	-	-	-	-	-	[76,102]
Quercetin *O*-glucoside-*O*-rutinoside II	-	3.67–132.7	-	-	-	-	-	[98]
Kaempferol	-	0.0028–10	0.061–19.65	0.0–3.72	5.60	-	12.1	[63,64,77,94,96,103]
Kaempferol 3-*O*-glucoside	-	0.024–1.36	0.008–6.01	-	3.40	-	0.30	[51,63,76,77,94,102,103]
Kaempferol 3-*O*-rutinoside	0.30–1.29	0.9–8.13	-	-	-	-	-	[54,76,90,102]
Kaempferol rutinoside-hexoside	-	0.13–1.08	-	-	-	-	-	[76]
Laricitrin	-	-	-	0.0–65.0 ^b^	-	-	-	[51]
Laricitrin 3-*O*-galactoside	-	-	-	41.0–710.0 ^b^	-	-	-	[51]
Laricitrin-3-O-glucuronide	-	-	-	151.0–640.0 ^b^	-	-	-	[51]
Laricitrin 3-*O*-glucoside	-	-	0.61–0.65	-	-	-	-	[81]
Isorhamnetin	-	0.0004–0.0024	-	-	-	-	-	[96]
Isorhamnetin 3-*O*-rutinoside	0.0–5.37	0.08–0.13	-	-	-	-	-	[54,74]
Isorhamnetin 3-*O*-glucoside	-	-	-	0.0–76.0 ^b^	-	-	-	[51]
Syringetin	-	-	-	0.0–119.0 ^b^	-	-	-	[51]
Syringetin 3-*O*-galactoside	-	-	-	70.0–742.0 ^b^	-	-	-	[51]
Syringetin 3-*O*-glucoside	-	-	0.77–0.97	85.0–594.0 ^b^	-	-	-	[51,81]
Syringetin 3-*O*-glucuronide	-	-	-	53.0–594.0 ^b^	-	-	-	[51]
Syringetin 3-*O*-rhamnoside	-	-	-	0.0–447.0 ^b^	-	-	-	[51]
Syringetin 3-*O*-pentoside	-	-	-	0.0–109.0 ^b^	-	-	-	[51]
**Flavan-3-ols**								
(+)-Catechin	-	0.13–84.34	0.067–81.8	0.13–387.48	4.0–77.0	0.38–1.44	-	[57,60,64,76,77,81,94,96,97,99,100]
(−)-Epicatechin	0.0–28.22	0.23–397.19	0.0014–20.70	0.0–129.51	13.60–369.0	1.17–1.54	-	[60,63,64,76,77,94,97,99,100,104]
Epigallocatechin	-	-	0.21–0.40	-	0.0–17.20	-	-	[94,99]
Epicatechin 3-gallate	-	0.29–3.12	0.48–19.27	-	-	-	-	[63,76]
Catechin glucoside	-	2.03–1.16	-	-	-	-	-	[76]
Procyanidin tetramer B type 1	-	0.33–1.01	-	-	-	-	-	[76]
Procyanidin tetramer B type 2	-	0.62–2.95	-	-	-	-	-	[76]
Procyanidin dimer B type 1	-	2.24–6.99	-	-	-	-	-	[76]
Procyanidin dimer B type 2	-	1.59–26.47	-	-	-	-	-	[76]
Procyanidin dimer B type 3	-	1.28–3.59	-	-	-	-	-	[76]
Procyanidin dimer B type 4	-	0.92–3.54	-	-	-	-	-	[76]
Procyanidin dimer B type 5	-	4.80–15.26	-	-	-	-	-	[76]
Propelargonidin dimer	-	0.29–0.77	-	-	-	-	-	[76]
Procyanidin pentamer B type	-	0.18–1.68	-	-	-	-	-	[76]
Dimer B2	-	-	0.40–1.51	-	-	-	-	[81]
Procyanidin B1	0.0–27.69	0.55–7.29	-	-	12.0–92.0	-	-	[57,74,100]
Procyanidin C1	0.0–8.6	-	-	-	-	-	-	[74]
**Flavanones**								
Naringenin	-	-	0.024–0.028	-	-	-	-	[94]
Naringenin hexoside	-	0.38–3.41	-	-	-	-	-	[76]
**Flavanonols**								
Taxifolin 3-*O*-rutinoside	-	0.43–100.52	-	-	-	-	-	[76]
Taxifolin *O*-deoxyhexosylhexoside	-	0.0–21.1	-	-	-	-	-	[90]
**Flavones**								
Luteolin	-	0.0027–0.020	-	-	-	-	-	[96]
Luteolin 7-*O*-glucoside	-	-	-	-	56.0	-	102.0	[103]
Apigenin 8-*C*-glucoside	-	-	0.057–0.14	-	-	-	-	[94]
**Coumarins**								
Aesculin	-	-	0.003–0.0011	-	-	-	-	[94]
**Chalcones**								
Phlorizin	-	-	0.041–0.57	-	-	-	-	[96]

TPC: total phenolic content; TAC: total anthocyanin content; GAE: gallic acid equivalents; C3G: cyanidin 3-*O*-glucoside equivalents; fw: fresh weight; t.r.: trace residues; *: g GAE/L; ^a^: mg/L; ^b^: mg per 100 g dry weight as equivalents of quercetin 3-*O*-glucoside; ^c^: relative content regarding total anthocyanins detected (%).

**Table 7 molecules-27-03294-t007:** Antidiabetic properties of cherries and blueberries.

Plant	Part Used/ Compound	Model	Description	Effects	References
**Enzymes inhibition studies**					
*Vaccinium myrtillus*	Fruits	In vitro	Evaluation of the inhibitory activity of anthocyanin-rich bilberry extract (BE) on *α*-glucosidase and *α*-amylase	↑ *α*-glycosidase and *α*-amylase inhibition in a mixed competitive manner.	[156]
*Vaccinium myrtillus*	Fruits	In vivo	Evaluation of the effect of BE on the digestive properties of carbohydrates in eight-week-old SPF-grade C57BL/6 J male mice	↓ Postprandial glucose	[156]
*Vaccinium corymbosum*	Fruits	In vitro	Evaluation of the highbush blueberries in the inhibition of *α*-glycosidase and *α*-amylase enzymes	↑ *α*-glycosidase and *α*-amylase inhibition	[157]
*Prunus avium*	Fruits	In vitro	Evaluation of the antidiabetic potential of hydroethanolic extract of sweet cherry	↑ *α*-glycosidase inhibition in a dose-dependent manner	[43,158]
*Prunus avium*	Stem, leaf, flower	In vitro	Evaluation of the antidiabetic potential of hydroethanolic extract and aqueous infusion of sweet cherry by-products	↑ *α*-glycosidase inhibition in a dose-dependent manner	[159]
*Vaccinium corymbosum*	Fruits	In vitro	Analysis of the inhibitory effect of phenolic compounds commonly present in berry on dipeptidyl-peptidase IV (DPP-IV)	Resveratrol and flavone are competitive inhibitors to (DPP-IV) Luteolin and apigenin bond to DPP-IV in a noncompetitive manner	[160]
**Pancreatic β-cells protection studies**					
n.a.	Resveratrol	In vitro	Evaluation of the effects of resveratrol on pancreatic *β*-cell function in mouse *β*-Min6 cells and human islets	↑ intracellular cAMP levels ↑ insulin secretion ↑ pancreatic *β*-cell function	[161]
*Prunus avium*	Fruits	In vivo	Evaluation of the effects of ethanolic extract on aloxan-induced diabetic rats	↓ blood glucose ↓ urinary microalbumin ↑ ctreatinine secretion level in urea	[162]
*Vaccinium myrtillus*	Leaf	In vivo	Analysis of the glucose homeostasis, pancreatic *β*-cell function, and insulin sensitivity in high-fat diet–induced in diabetic male C57BL/6J mice	↓ plasma glucose ↓ glycated hemoglobin ↓ insulin resistance ↑ mRNA levels of pancreatic *β*-cell ↑ pancreatic insulin signaling ↓ transcriptional expression of the *β*-cellimproved insulin sensitivity ↑ insulin signaling	[163]
*Vaccinium myrtillus*	Fruits	Clinical trial	Evaluation of the effects of purified anthocyanins on dyslipidemia, oxidative status, and insulin sensitivity in patients with type 2 diabetes	improved dyslipidemia ↑ antioxidant status ↓ plasma glucose ↓ insulin resistance ↓ LDL cholesterol and triglycerides ↑ HDL	[164]
**Insulin release and regulation**					
*Prunus avium*	Fruits	In vitro	Evaluation of three different phenolic fractions (anthocyanins-rich fraction (ARF), hydroxycinnamic acids-rich fraction (HRF) and flavonols-rich fraction (FRF)) in glucose consumption by HepG2 cells	↑ glucose consumption ↑ insulin sensitivity inhibited excessive gluconeogenesis	[165]
*Vaccinium myrtillus*	Fruits	In vivo	Analysis of blueberry effects glucose metabolism and pancreatic *β*-cell proliferation in high fat diet (HFD)-induced obese mice.	↑ insulin sensitivity ↑ glucose tolerance	[166]
*Vaccinium ashei* *Vaccinium corymbosum*	Fruits	Clinical trial	Evaluation of the effect of daily dietary supplementation with bioactives from blueberries on whole-body insulin sensitivity in men and women	↑ insulin sensitivity	[167]
n.a.	Resveratrol	In vitro	Evaluation of the protective effects of resveratrol in *β*-cell dysfunction in INS-1 cells	↑ glucose-stimulated insulin secretion ↑ SIRT1 expression restored the function of INS-1 cell	[168]
*Vaccinium myrtillus*	Fruits	In vivo	Analysis of the otective effects of blueberry anthocyanin extract (BAE) against oxidative stress and the roles of SIRT1 and NF-κB	↑ SIRT1 expression ↑ SOD and GSH activity	[169]
*Vaccinium myrtillus*	Fruits	In vitro	Evaluation of the role of berry phenolic compounds to modulate incretin-cleaving DPP-IV and its substrate glucagon-like peptide-1 (GLP-1), insulin secretion	↑ insulin secretion Upregulated expression of mRNA of insulin-receptor	[170]

↑—increase; ↓ decrease.

## Data Availability

Not applicable.

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
