# Peer review of "Cherries and Blueberries-Based Beverages: Functional Foods with Antidiabetic and Immune Booster Properties"

_molecules, 2022, doi:10.3390/molecules27103294_

Round 1
Reviewer 1 Report
Dear Authors
The manuscript is interesting and deserves cosideration in this journal, the topic is within the aim and scope and the text is correctly organized. However I found the following concerns about it:
- Carefully revise the english for typos and errors
- improve figures resolution, at least 300 dpi
- the review should include also alternative plant baverages based of recent formulation, such as low fermentation beer, at this regard I suggest to add a brief paragraph on it. Some example of literature are reported below: "Artisanal fortified beers: Brewing, enrichment, HPLC-DAD analysis and preliminary screening of antioxidant and enzymatic inhibitory activities", "Chemical profiling, antioxidant, enzyme inhibitory and molecular modelling studies on the leaves and stem bark extracts of three African medicinal plants", "An overview on plants cannabinoids endorsed with cardiovascular effects".
Author Response
Response to Reviewers’ Comments (Manuscript molecules-1725093)
Ms. Ref. No. molecules-1725093
Title: Cherries and blueberries-based beverages: functional foods with antidiabetic and immune booster properties
Reviewer 1
Authors’ response: First of all, we would like to thank your kind comments and compliments made on our manuscript. Following the comments received, the changes made by us are highlighted in the revised version, in accordance with the request. In addition, English was all revised. Even so, after each Reviewer’s comment, we indicated the main changes introduced and the corresponding lines. Thank you.
- Carefully revise the english for typos and errors
Authors’ response: Thank you so much for your commentary. As suggested, the English was all revised (please see now the revised version).
- improve figures resolution, at least 300 dpi
Authors’ response: Thank you so much for your commentary. As suggested, the figures’ resolution was improved (please see now the revised version).
- the review should include also alternative plant baverages based of recent formulation, such as low fermentation beer, at this regard I suggest to add a brief paragraph on it. Some example of literature are reported below: "Artisanal fortified beers: Brewing, enrichment, HPLC-DAD analysis and preliminary screening of antioxidant and enzymatic inhibitory activities", "Chemical profiling, antioxidant, enzyme inhibitory and molecular modelling studies on the leaves and stem bark extracts of three African medicinal plants", "An overview on plants cannabinoids endorsed with cardiovascular effects".
Authors’ response: Thank you so much for your recommendation. As suggested, the same was added (please see now the references of the revised version and lines 88 to 90 of the revised version).
Author Response
Response to Reviewers’ Comments (Manuscript molecules-1725093)
Ms. Ref. No. molecules-1725093
Title: Cherries and blueberries-based beverages: functional foods with antidiabetic and immune booster properties
Reviewer 2
In this work, the review reinforces the idea that cherries and blueberries can be incorporated into new pharmaceutical products, smart foods, functional beverages, and nutraceuticals and be effective in preventing and/or treating diseases mediated by inflammatory mediators, reactive species and free radicals. In addition, the prospects and possible directions are discussed as well to provide further insight into the future development of this field. These results are desirable for developing functional foods with antidiabetic and immune booster properties in term of the novelty. In my opinion, this review would benefit the readers who are interested in the relevant field. I recommend the publication of this manuscript on Molecules addressing following revisions.
Authors’ response: First of all, we would like to thank your kind comments and compliments made on our manuscript. Following the comments received, the changes made by us are highlighted in the revised version, in accordance with the request. In addition, English was all revised. Even so, after each Reviewer’s comment, we indicated the main changes introduced and the corresponding lines. Thank you.
- Some chapters (for example: section 1, 2) had better give relevant or more representative pictures to enrich the article and make it easier for readers to understand at a glance.
Authors’ response: Thank you so much for your suggestion. However, and owning to the time that the editor made available to us to make all revisions, it is impossible for us to add more pictures. I hope the reviewer understands, thank you so much (please see now the revised version).
- The authors should check the format of reference carefully (for example: Ref.223, 224).
Authors’ response: Thank you so much for your commentary. All references were revised and corrected, thank you so much (please see now the revised version).
- Picture and table formats need to be further unified or improved.
Authors’ response: Thank you so much for your commentary. As suggested, figures and table 6 were improved (regarding the other ones, after discussion, the authors decided to them to keep them as they are because they think that, in this way, everything is more clarified, I hope the reviewer understands it, thank you so much) (please see now the revised version).